# Genome-scale model of *Rothia mucilaginosa* predicts gene essentialities and reveals metabolic capabilities

Nantia Leonidou,[1,2,3,4,5] Lisa Ostyn,[6] Tom Coenye,[6] Aurélie Crabbé,[6] Andreas Dräger[1,4,7]

**ABSTRACT**  Cystic fibrosis (CF), an inherited genetic disorder caused by mutations in the cystic fibrosis transmembrane conductance regulator gene, results in sticky and thick mucosal fluids. This environment facilitates the colonization of various microorganisms, some of which can cause acute and chronic lung infections, while others may positively impact the disease. *Rothia mucilaginosa*, an oral commensal, is relatively abundant in the lungs of CF patients. Recent studies have unveiled its anti-inflammatory properties using *in vitro* three-dimensional lung epithelial cell cultures and *in vivo* mouse models relevant to chronic lung diseases. Apart from this, *R. mucilaginosa* has been associated with severe infections. However, its metabolic capabilities and genotype-phenotype relationships remain largely unknown. To gain insights into its cellular metabolism and genetic content, we developed the first manually curated genome-scale metabolic model, *i*RM23NL. Through growth kinetics and high-throughput phenotypic microarray testings, we defined its complete catabolic phenome. Subsequently, we assessed the model's effectiveness in accurately predicting growth behaviors and utilizing multiple substrates. We used constraint-based modeling techniques to formulate novel hypotheses that could expedite the development of antimicrobial strategies. More specifically, we detected putative essential genes and assessed their effect on metabolism under varying nutritional conditions. These predictions could offer novel potential antimicrobial targets without laborious large-scale screening of knockouts and mutant transposon libraries. Overall, *i*RM23NL demonstrates a solid capability to predict cellular phenotypes and holds immense potential as a valuable resource for accurate predictions in advancing antimicrobial therapies. Moreover, it can guide metabolic engineering to tailor *R. mucilaginosa*'s metabolism for desired performance.

**IMPORTANCE**  Cystic fibrosis (CF) is a genetic disorder characterized by thick mucosal secretions, leading to chronic lung infections. *Rothia mucilaginosa* is a common bacterium found in various parts of the human body, acting as a normal part of the flora. In people with weakened immune systems, it can become an opportunistic pathogen, while it is prevalent and active in CF airways. Recent studies have highlighted its anti-inflammatory properties in the lower pulmonary system, indicating the intricate relationship between microbes and human health. Herein, we have developed the first manually curated metabolic model of *R. mucilaginosa*. Our study examined the previously unknown relationships between the bacterium's genotype and phenotype and identified essential genes that impact the metabolism under various conditions. With this, we opt for paving the way for developing new strategies in antimicrobial therapy and metabolic engineering, leading to enhanced therapeutic outcomes in cystic fibrosis and related conditions.

**KEYWORDS**  *i*RM23NL, *Rothia mucilaginosa* DSM20746, ATCC 25296, constraint-based modeling, flux balance analysis, flux variability analysis, mathematical network, genome-

Address correspondence to Nantia Leonidou, nantia.leonidou@uni-tuebingen.de.

The authors declare no conflict of interest.

See the funding table on p. 21.

scale metabolic model, metabolic engineering, pathway analysis, SBML, Gram-positive, nasal microbiome, lung infections, cystic fibrosis, antimicrobial strategies

*R*othia mucilaginosa is a Gram-positive, encapsulated, non-motile, and non-spore-forming bacterium of the *Micrococcaceae* family (1, 2). While it is mainly aerobic, it may also grow anaerobically as it can switch to fermentation or other non-oxygen-involving pathways. *R. mucilaginosa* is a common commensal of the normal oral, upper and lower respiratory tract, and part of the skin florae in humans (1, 3–6). This means it coexists harmlessly within the host and may even provide benefits. Nonetheless, it can also act as an opportunistic pathogen, particularly in individuals with weakened immune systems, as an etiological agent of serious infections such as endocarditis, sepsis, and meningitis (7). Janek et al. highlighted the high prevalence of *R. mucilaginosa* within the nasal microbiome (8). Moreover, they report its susceptibility to certain staphylococcal bacteriocins, indicating its major competition with the nasal staphylococci and the substantial impact of bacteriocins in shaping the nasal microbiota. In 2020, Uranga et al. revealed that *R. mucilaginosa* produces the strongest $Fe^{3+}$-binding archetypal siderophore known, called enterobactin (9). This attribute contributes to its competition with oral microbiota (the cariogenic *S. mutans*, *A. timonensis*, and *Streptococcus* sp.) as well as four methicillin-resistant strains of *S. aureus* (MRSA). Enterobactin is a type of siderophore produced by bacteria to scavenge, chelate, and transport ferric irons from their surrounding environment. These are essential for bacteria when iron is scarce as they facilitate their acquisition necessary for their growth and metabolic processes.

Prior metagenomic sequencing analyses have unveiled the prevalence of *R. mucilaginosa* at high abundances and its enhanced metabolic activity in the lungs of cystic fibrosis (CF) patients (10, 11). CF is caused by the hereditary mutation of the cystic fibrosis transmembrane conductance regulator (CFTR) gene that disrupts the transepithelial movement of ions, leading to an aberrant accumulation of thick and sticky mucus within the airways. The impaired immune clearance creates a hypoxic environment (12) promoting the polymicrobial colonization of opportunistic microbes together with fungi and viruses, ultimately resulting in persistent and recurring infections (13). Guss et al. and Bittar et al. declared *R. mucilaginosa* as an emerging CF bacterium (14, 15), while Lim et al. provided evidence supporting that *R. mucilaginosa* is a frequently encountered and metabolically active inhabitant of CF airways (16). Additionally, a study from 2018 shows that the opportunistic pathogen *Pseudomonas aeruginosa*, which frequently causes infections in CF patients, builds essential primary metabolites, like glutamate, by utilizing compounds produced by *R. mucilaginosa* (17). This symbiotic interaction implies that *P. aeruginosa* benefits from its neighboring microbes, which contributes to its pathogenesis in the CF lungs. On the other hand, Rigauts et al. revealed the anti-inflammatory activity of *R. mucilaginosa* in the lower respiratory tract, which could impact the seriousness of chronic lung diseases (18).

In systems biology, genome-scale metabolic models (GEMs) represent comprehensive reconstructions of organisms' metabolic networks. They are built using genomic sequences and comprise all known biochemical reactions and associated genes. These models provide systems-level insights into cellular metabolism, allowing researchers to simulate and analyze the flow of metabolites through these networks (19). The interactions among reactions and metabolites in a metabolic model are mathematically represented with a stoichiometric matrix (20). In the past years, an array of *in silico* methods has been developed to analyze GEMs and derive valuable hypotheses. Flux balance analysis (FBA) is such a powerful computational technique that operates on the principle of achieving a steady state by optimizing the flux (rate) of metabolites through reactions while accounting for various constraints such as stoichiometry, thermodynamics, and uptake/secretion boundaries (21). Applying flux balance analysis on a GEM provides insights into the intricate biological system interactions. This analytical approach facilitates the prediction of cellular phenotypes and identification of promising drug targets and contributes to optimizing biotechnological processes (22). Moreover,

such models can guide genetic engineering by suggesting genetic modifications that could enhance desired product formation or cellular behavior. Further applications include ameliorating culture media by incorporating components that increase bacterial growth rates. So far, GEMs have been an invaluable resource in the systems biology field that helped untangle the metabolism of various organisms and especially of high-threat pathogens (23, 24). As described above, *R. mucilaginosa* has gained great interest in the context of polymicrobial CF environments. However, its metabolic capabilities and genotype-phenotype relationships in isolated monoculture settings remain largely unexplored.

Here, we present the first manually curated and high-quality GEM of *R. mucilaginosa*, *i*RM23NL, striving to understand its metabolism and unique phenotypes under diverse conditions. Our simulation-ready network accounts for thousands of reactions and is available in a standardized format following the community guidelines (25). Through growth kinetic experiments and high-throughput phenotypic microarray assays, we validated *i*RM23NL's accuracy in predicting growth and substrate utilization patterns. We refined the reconstruction by comparing the *in vitro* results to *in silico* simulations, resulting in novel metabolic reactions and genes. To our knowledge, this is the first study presenting high-throughput nutrient utilization and comprehensive growth data for *R. mucilaginosa*. Finally, we employed FBA to formulate novel gene essentiality hypotheses that could expedite the development of antimicrobial strategies. Figure 1 summarizes the experimental and computational work presented here.

## RESULTS

### Reconstruction of a high-quality metabolic model for *R. mucilaginosa* DSM20746

The pipeline we previously developed (26) was used to build the first high-quality and manually curated GEM of *R. mucilaginosa* DSM20746 (ATCC 25296). An initial draft metabolic model was derived with CarveMe (27) and is based on the Biochemical, Genetical, and Genomical (BiGG) identifiers (28). The translated sequence with over 1,700 proteins and the Gram-positive-specific template were employed. This enabled us to build a more precise reconstruction considering information on the peptidoglycan layer for the biomass objective function (BOF). The draft network contained 1,015 reactions (141 pseudo-reactions), 788 metabolites, and 265 genes (Fig. 2). In the first gap-filling stage (Draft_2), we expanded the list of reactions based on the annotated genome and growth kinetics data in diverse growth environments. For this, we extensively indexed organism-specific literature and databases and included additional enzymatic reactions together with 50 new gene-protein-reaction associations (GPRs). Subsequently, high-throughput nutrient utilization assays and model validation incorporated further reactions and their associated metabolic genes. Non-metabolic genes, which take part in other cellular processes, e.g., signaling pathways or transcription, were not considered. In total, 95 reactions, together with their associated metabolites, were newly added into the model, along with 121 novel GPRs, increasing the genetic coverage. Over 60% of the transport reactions have a GPR assigned, while 63% of the total enzymatic reactions have at least one gene assigned. Moreover, missing exchange reactions were added to all extracellular metabolites to represent the exchange of substrates between the extracellular environment and the model. The strain-specific BioCyc31 database was further employed to correct the reversibility of biochemical reactions, while duplicated reactions and metabolites were eliminated. In all cases, when no organism-specific information was available, we leveraged data from closely related species based on our phylogenomic analysis (Fig. 3). According to the calculated average nucleotide identity (ANI) matrix, *R. mucilaginosa* exhibits a similarity to six out of the 12 tested *Rothia* genomes. More specifically, it shares a greater resemblance with *R. aeria* and *R. dentocariosa* underscoring a closer evolutionary relationship between these species.

*R. mucilaginosa* is primarily aerobic, efficiently generating ATP through oxic respiration; however, in low-oxygen or oxygen-absent conditions, it shifts to anaerobic

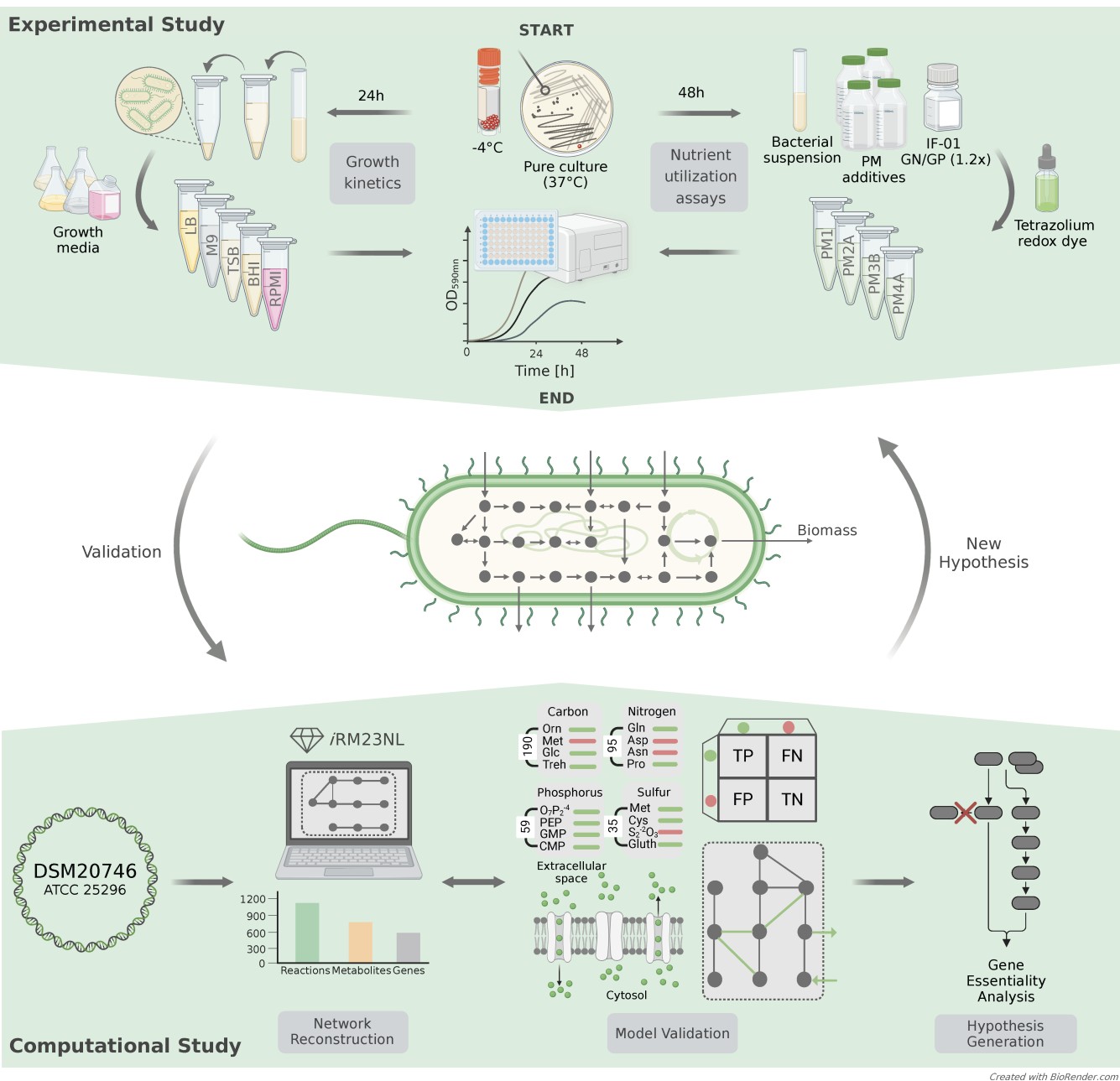

**FIG 1** Construction and validation flowchart of the metabolic network for *R. mucilaginosa*, *i*RM23NL. The study is divided into the experimental and computational phases. The proteome-derived metabolic reconstruction and curation was done based on the workflow we described elsewhere (26).

metabolism to produce energy. This metabolic adaptability enables *R. mucilaginosa* to adapt in microaerophilic environments like the oxygen-restricted conditions in CF lungs (16). Our draft model lacked the ability to demonstrate anaerobic growth. Therefore, we investigated the metabolic cascade and systematically incorporated missing enzymes to ensure that the model can simulate growth even in the absence of oxygen by identifying and integrating alternative pathways. This refinement included the incorporation of enzymatic reactions, such as the superoxide dismutase (SPODM) and catalase (CAT) that are responsible for the breakdown of radical reactive oxygen species (ROS) and shielding the cell against oxidative damage (Fig. 4 Panel A). Such scavenging enzymes play an integral role in counteracting the harmful effects of ROS during anaerobic respiration (31). However, during this process, we found no associated GPRs for CAT within the

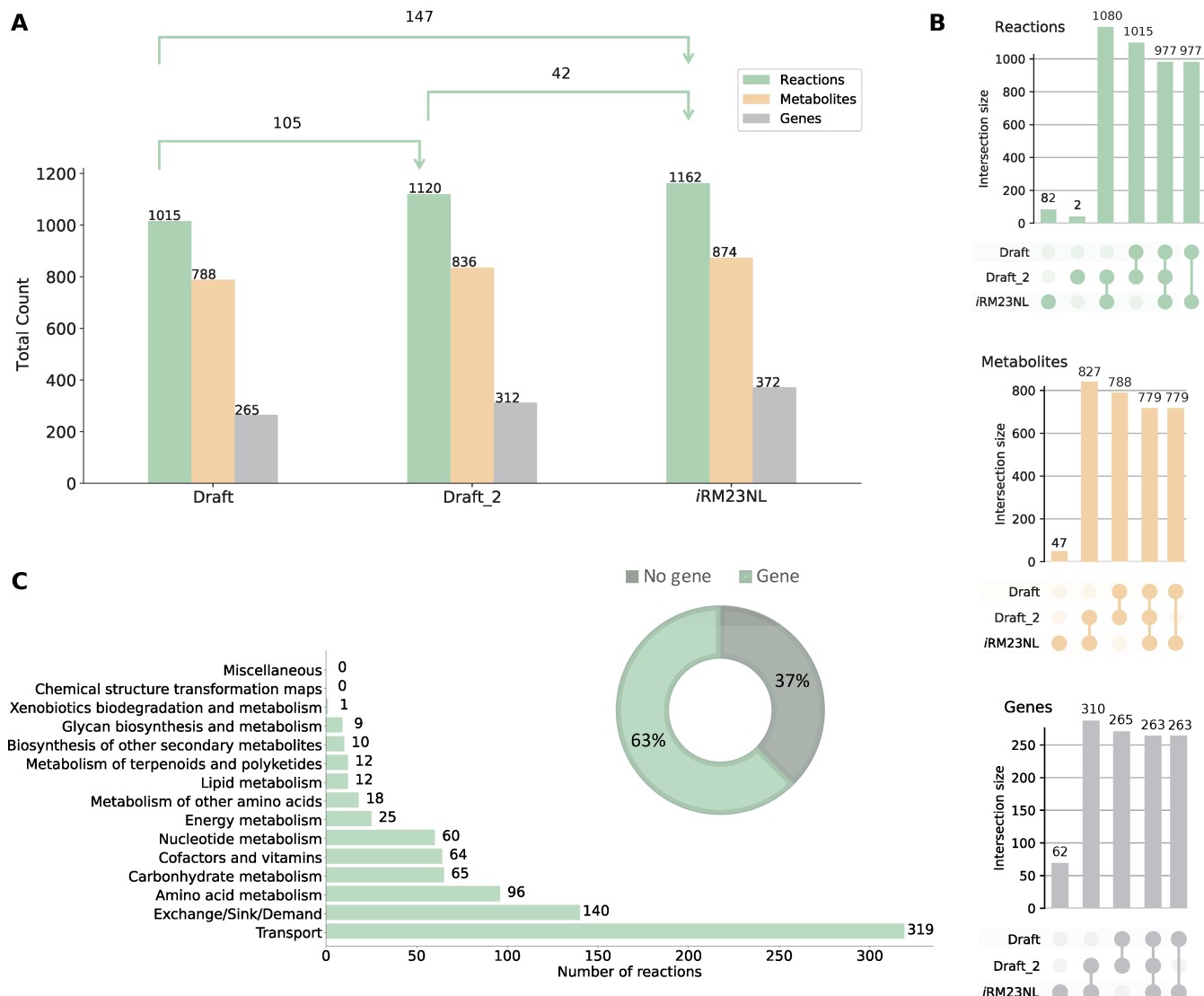

**FIG 2** Properties of the genome-scale metabolic model for *R. mucilaginosa* DSM20746, *i*RM23NL. (A) Evolution of metabolic network content from its initial draft to the final stage of extensive manual gap-filling. The shifts in the sets' sizes are also displayed in each stage. The first stage of gap-filling is denoted by Draft_2, while the final stage is upon validation with experimental data. (B) UpSet plots comparing sets between three model versions created using the UpSetPlot package (29). The numbers indicate the cardinality of the respective set. (C) Subsystem-level statistics within pathways along with the distribution of gene- and non-gene-associated reactions. The pathway analysis was limited to reaction identifiers that could be successfully mapped to Kyoto Encyclopedia of Genes and Genomes (KEGG) (30) reactions.

organism-specific BioCyc database. Additional scavenging enzymes like glutathione and thioredoxin reductases essential for maintaining the redox balance (32) were already present in the initial draft model (GTHOr, GTHRDabc2pp, and TRDR). Altogether, the final model, *i*RM23NL, contains 1,162 reactions (619 gene-associated; 65 catalyzed by enzyme complexes, 70 catalyzed by isozymes, and 484 by simple gene association), 171 exchange and sink reactions, 874 metabolites (558 in cytoplasm, 148 in periplasm, and 168 in the extracellular space), and 372 genes (Fig. 2). The model's metabolic coverage is at 3.12%, which indicates a high level of modeling detail regarding reactions, enzymes, and their associated genes (33). Additionally, we enriched the model elements with numerous database cross-references (34), while appropriate and precise Systems Biology Ontology (SBO) terms were assigned to each model entity using the SBOannotator package (35). The presence of no energy-generating cycles (EGCs) was

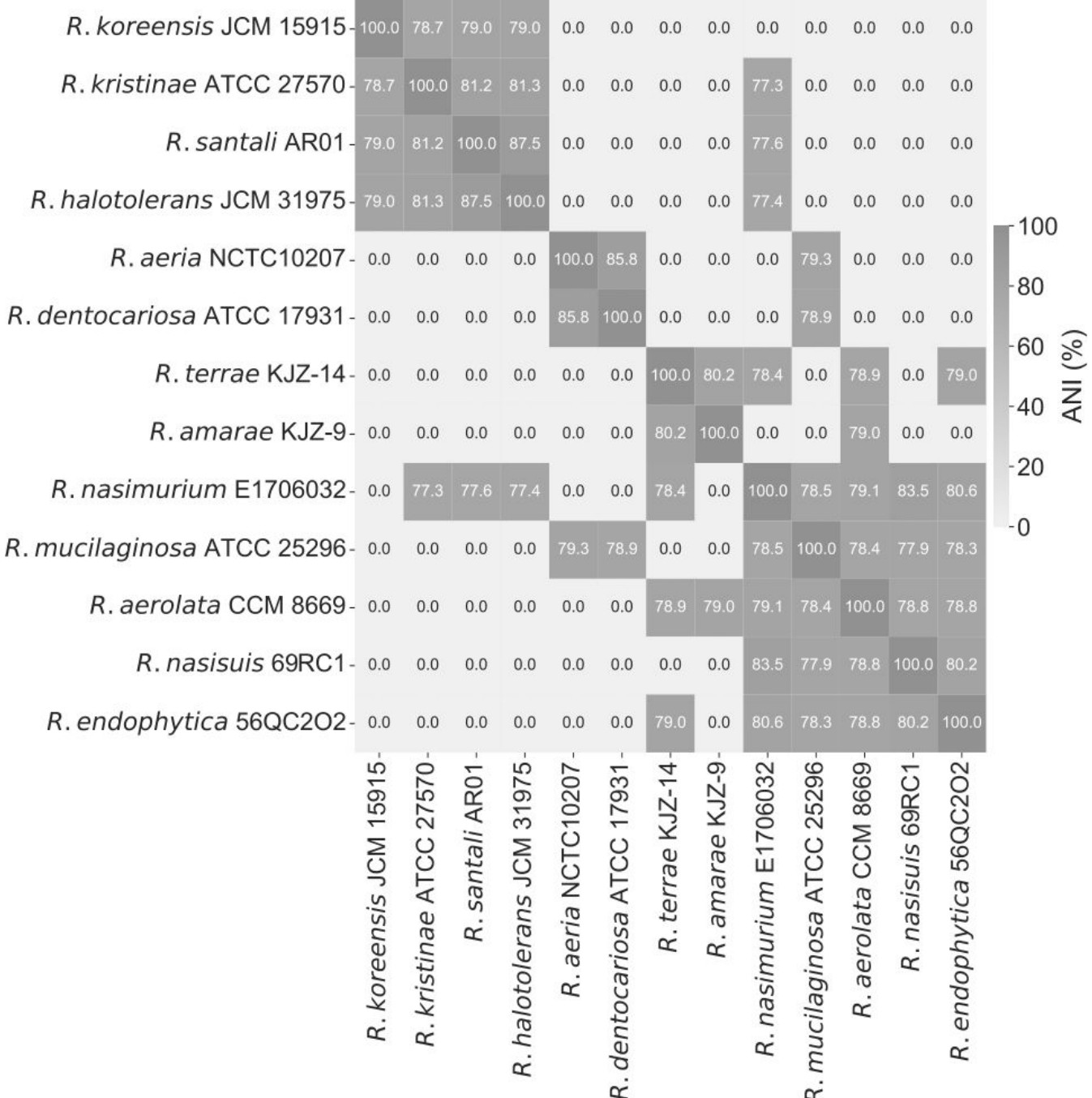

**FIG 3** Phylogenomic all-vs-all analysis between 13 *Rothia* species. Based on the calculated ANI matrix, *R. mucilaginosa* is mostly similar to six out of 13 genomes, with higher similarity to *R. aeria* and *R. dentocariosa*.

ensured and controlled after each curation stage, and the mass- and charge-imbalances were corrected. With this, the final Metabolic Model Testing (MEMOTE) (36) score of *i*RM23NL is 89%, while with highly specific SBO terms, the score drops by 2%. The final curated model, *i*RM23NL, is available as a supplementary file in Systems Biology Markup Language (SBML) Level 3 Version 1 (37) and JavaScript Object Notation (JSON) formats with the flux balance constraints (fbc) and groups plugins available.

The first validation step of *i*RM23NL aimed to evaluate its ability to correctly simulate biomass production across diverse environmental conditions and growth media

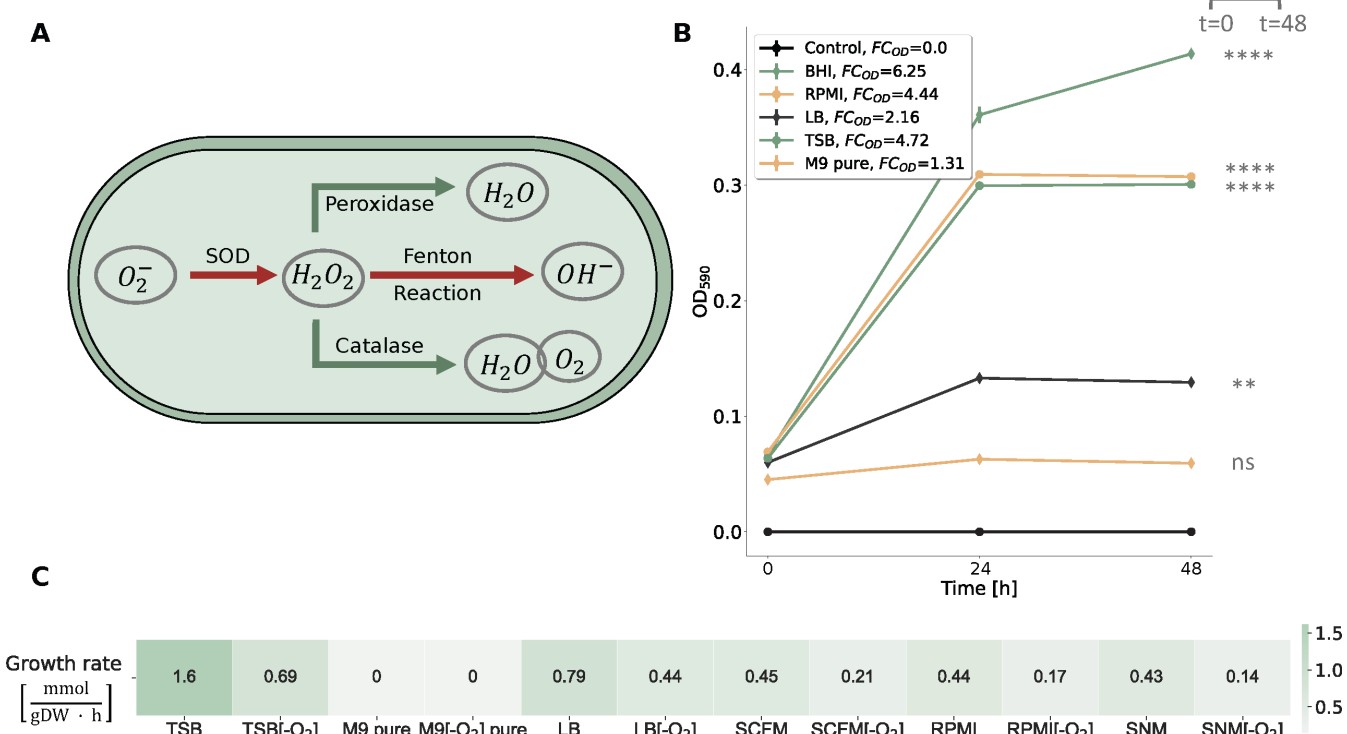

**FIG 4** Investigation of *R. mucilaginosa*'s growth behavior in different nutrient media. (A) Metabolic response of *R. mucilaginosa* under anaerobic stress as represented in *i*RM23NL. Reduction process of oxygen ($O_2^-$) generating ROS is indicated by red arrows, while pathways highlighted in green arrows represent reactions governed by ROS scavenging enzymes leading to bacterial cell detoxification. (B) Experimentally-derived growth curves for *R. mucilaginosa* DSM20746 in multiple liquid growth media along with the respective fold changes (FCs) of the acquired optical densities (ODs) at 590 nm, as defined in Equation 1. The data shown here are an average of three biological replicates ($n = 3$). Based on the experimental results, a threshold of $FC_{OD} = 1.4$ was established to qualitatively describe bacterial growth. We verified the correctness of the threshold by performing statistical analysis as described in Materials and Methods. All data are normally distributed, while there is no significant difference between their variances. The asterisks flag the significance levels. The BHI medium was used as a baseline, while the Control line represents blank measurements of pure media. Bacterial growth was aerobically measured by the OD at 590 nm (ordinate) at three distinct time points ranging from 0 h to 48 h (abscissa). (C) *In silico*-simulated growth rates using *i*RM23NL. Detailed *in silico* media formulations are provided in Table S2.

formulations. To elucidate the bacterium's optimal conditions and metabolic preferences, we experimentally tested five commonly used media, including three general nutrient media; brain heart infusion (BHI) and Luria-Bertani (LB), and tryptic soy broth (TSB), and two defined media; M9 minimal medium (M9) pure and Roswell Park Memorial Institute (RPMI) (Fig. 4 Panel B). The BHI medium was used as a baseline for the *in vitro* experiments since it is a known and well-established environment for the growth of *R. mucilaginosa* and enabled us to compare the bacterium's growth characteristics to the newly tested media. For the *in silico* simulations, we applied FBA and added additional constraints to the linear programming problem defined in Equation 5. In more detail, we specified the flux constraints such that only extracellular metabolites defined in the medium of interest could flow freely through the system (unconstrained, finite fluxes), while the remaining fluxes were constrained to zero. We compared the *in vitro* to the *in silico* observed growth using the $FC_{OD}$ as a qualitative measure of growth (see Materials and Methods). Furthermore, we compared the OD at the start and the end of the experiment, considering a statistically significant difference between these measurements as an indication of growth. Our metabolic network, *i*RM23NL, simulated positive fluxes through the biomass reaction for all tested media except for the M9 pure medium, where a zero flux was observed. These findings align with the experimentally observed data. More specifically, there is no statistically significant difference in OD between the initial and final time-points in M9 pure medium (*P*-value = 0.1202 and

$FC_{OD} < 1.4$) indicating no significant growth. Conversely, in the remaining examined media, statistically significant growth was observed (P-value = 0.00006–0.00142 and $FC_{OD} > 1.4$) indicating significant growth in these settings. The highest aerobic growth rate was predicted in TSB [1.6 mmol/($g_{DW} \cdot$ h)], while the lowest biomass production flux was recorded for the M9 pure medium containing only essential salts. However, the RPMI medium followed as the second-highest in supporting bacterial *in vitro* cellular growth, offering a defined medium suitable for *R. mucilaginosa*'s cultivation. Although *R. mucilaginosa* increased its biomass after 24 h, it slightly declined after 48 h. On the other hand, the simulated network resulted in a contrary outcome compared to the expected experimental effect. More specifically, *i*RM23NL simulated a lower flux through biomass [0.44 mmol/($g_{DW} \cdot$ h)] with RPMI when compared to LB. It is important to note here that in order to simulate growth in RPMI medium, six metal ions [cobalt ($Co^{2+}$), copper ($Cu^{2+}$), manganese ($Mn^{2+}$), zinc ($Zn^{2+}$), ferric iron ($Fe^{3+}$), and ferrous iron ($Fe^{2+}$)] were supplemented. These compounds were missing from the providers' medium formulation. Our findings underscored *R. mucilaginosa*'s adaptability to various nutritional environments, growing best in nutrient-rich conditions while revealing specific growth requirements beyond minimal settings.

We further employed *i*RM23NL to examine whether it could generate biomass within the human nasal environment and the CF lungs. For this purpose, we performed *in silico* simulations using the synthetic cystic fibrosis sputum medium (SCFM) (38) and synthetic nasal medium (SNM) (39) (Fig. 4 Panel C). Our computational model successfully simulated positive growth in both media, with a growth rate of 0.43 mmol/($g_{DW} \cdot$ h) in SNM and 0.45 mmol/($g_{DW} \cdot$ h) in SCFM. These results align with the documented metabolic activity of *R. mucilaginosa* in CF lungs and its frequent isolation from the human nasal cavity. Notably, the observed growth rates closely resembled the flux rate predicted for biomass production in RPMI medium. Additionally, we confirmed that *i*RM23NL accurately represented *R. mucilaginosa*'s capacity for facultative anaerobic respiration. In more detail, when the oxygen uptake was turned off *i*RM23NL could successfully exhibit growth using alternative metabolic pathways across all tested nutritional media. When the oxygen level was decreased, the model predicted up to 68% reduction in biomass yield compared to aerobic conditions. Consequently, the remarkably lower anaerobic rates in all tested media mimic *R. mucilaginosa*'s inherent facultative anaerobic capabilities.

## Nutrient utilization profile of *R. mucilaginosa* and predictive performance of *i*RM23NL

We experimentally characterized the metabolic phenotype of *R. mucilaginosa* DSM20746 using four 96-well Biolog PM microplates (Fig. 5). These high-throughput assays serve as proxies for bacterial growth by measuring cellular respiration across several conditions. Active respiration in the minimal medium is detected by the reduction of tetrazolium dye over time, indicating the utilization of the provided sole source (40). We cultivated our strain in a minimal medium supplemented with various sources, and growth was monitored over 48 h to identify suitable nutrients for the bacterium (as described in Materials and Methods). The derived OD measurements were normalized according to the average growth over replicates per plate and converted to qualitative data representing non-growth (NG) or growth (G). In total, we tested the uptake and utilization of 379 distinct carbon, nitrogen, phosphorus, and sulfur substrates. *R. mucilaginosa* demonstrated the ability to utilize 61 of 190 tested carbon substrates, including carboxylates, saccharides, and amino acids, while 10 of 95 were found to be viable nitrogen sources (Fig. 5 Panel B). Furthermore, out of 59 tested phosphorus sources, *R. mucilaginosa* exhibited a loss of metabolic activity for 28 compounds, resulting in a non-viable phenotype, while only 71.4% of all analyzed sulfur substrates supported positive growth. More specifically, 6 inorganic phosphorus (IP), 14 organic phosphorus (OP), 2 cyclic nucleoside monophosphates (cNMPs), and 9 nucleoside monophosphates (NMPs) were successfully utilized as sole phosphorus sources (Fig. 5, Panel C). The experimentally

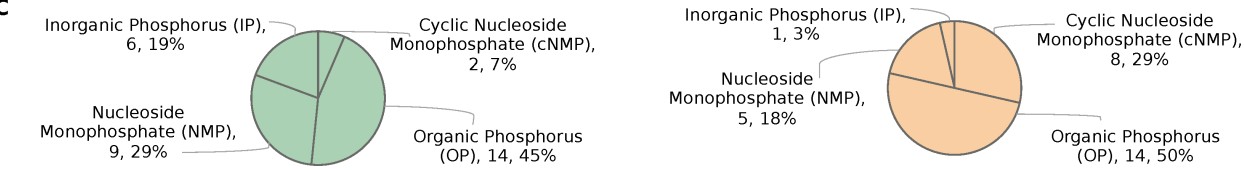

**FIG 5** Complete experimentally-derived nutrient utilization phenome of *R. mucilaginosa* DSM20746. (A) Utilization of individual nutrients by the bacterium across four Biolog phenotypic microarrays. Bacterial growth was measured by OD at 590 nm. (B) Numerical summary nutrient sources experimentally tested in each Biolog phenotype microarray (PM), classified into those resulting in bacterial growth and those that *R. mucilaginosa* could not utilize. (C) Categorization of all tested phosphorous sources during the high-throughput Biolog assay. Utilization of totally 31 phosphorus sources resulted in positive phenotype (green chart), while the cell exhibited an inability to utilize the remaining 28 (orange chart).

defined nutrient utilization phenome of *R. mucilaginosa* can be found in Fig. S1. An overview of all experimentally tested substrates, along with the assay results, can be

found in Table S3. We independently confirmed the Biolog nutrient utilization data by testing the ability of DSM20746 to grow on minimal media in the presence of 10 compounds (see Materials and Methods, Fig. S2).

Additionally, we evaluated the predictive performance of our metabolic model by using various C-, N-, P-, and S-containing substrates. All compounds from the high-throughput phenotypic data were mapped to BiGG (28) identifiers and subsequently to *i*RM23NL. In total, 286 could be successfully mapped to the BiGG database. From these, 126 existed as extracellular metabolites in *i*RM23NL and were considered for further analysis. Model simulations were performed under aerobic conditions with the minimal medium defined in Table S2 and FBA (see Materials and Methods). An extracellular reaction was enabled for each tested substrate to force the model to use its transporters. Discrepancies between the Biolog data and the model simulations were utilized as basis for hypotheses to further improve and refine the network reconstruction. We resolved most inconsistencies via extensive literature mining and iterative gap analysis. For this, we used the organism- and strain-specific BioCyc (33) database. Throughout this process, we encountered different scenarios regarding incorrect model predictions. These included compounds present in all compartments, including the extracellular space, as well as substrates defined within the intracellular space and periplasm, with no transporter defined toward the extracellular space. If the experimental results indicated utilization of an undefined compound, we searched BioCyc (33) to find strain-specific and gene-based missing transporters or enzymatic reactions. When no organism-specific evidence was available, we sought supporting data from genomically identical species (Fig. 3). For instance, the compound 3-sulfino-L-alanine (3sala) was initially absent from any compartment in the preliminary draft model. Since no strain-specific information was available, we conducted a homology-based search using Basic Local Alignment Search Tool (BLAST) (41) to find genes with high similarity (similarity threshold: > 80%) in related species. Subsequently, we identified cysteine desulfurase (SULFCYS) along with three associated transport reactions (proton-mediated; SULFCYSpp, diffusion; SULFCYStex, and ABC transport; SULFCYSabc) that displayed a similarity over 80% with *R. dentocariosa*. These components were consequently incorporated into *i*RM23NL, resulting in the expected positive utilization phenotype. Generally, false negative or false positive predictions arise from missing or erroneous involvement of transporters, respectively. We resolved false positives by removing transport reactions lacking supporting gene evidence or adjusting their reversibility to facilitate export solely. More specifically, initial model predictions indicated that *i*RM23NL could not sustain growth when supplied with either L-cysteate (Lcyst) or AMP (amp) as sole sources, while Biolog assays indicated the opposite. To rectify this, we introduced the corresponding irreversible transporters (LCYStex and AMPt) and enabled their *in silico* utilization of these compounds. Moreover, several metabolites (e.g., phosphoenolpyruvate; pep, trimetaphosphate; tmp, hypotaurine; hyptaur, and inorganic triphosphate; pppi) which were absent from the initial draft model but exhibited positive growth in utilization assays were subsequently incorporated into the final network, leading to additional true positives predictions. All in all, over 50 transport reactions were added into the network, while 37 wrongly added enzymatic functions were removed. We also incorporated novel GPRs encoding over 60 biochemical reactions. Nevertheless, we identified approximately 20 instances where the resolution of inconsistencies necessitated the inclusion of metabolic reactions lacking associated gene evidence. For instance, to enable the utilization of L-aspartate, we introduced a transporter via diffusion from extracellular to periplasm (ASPtex), for which no associated GPR was available. Similar scenarios arose for other compounds, e.g., D-galactose, D-glucuronate, and acetate. These instances underscore knowledge gaps in the metabolism of DSM20746 that require in-depth investigation. In total, 14 carbon and nitrogen sources failed to promote growth in *i*RM23NL. Surprisingly, all of these sources had corresponding transport reactions *i*RM23NL but still remained ineffective (e.g., L-fucose, L-arabinose, and L-rhamnose) and

nitrogen (L-tyrosin). We could not find further information on their transport or metabolic mechanism either in the genome annotation or the literature.

In summary, the final prediction accuracy of nutrient assimilation and utilization achieved by *i*RM23NL was 77% for carbon sources (MCC for PM1 = 0.52 and PM2A = 0.58), 94.4% for nitrogen sources (MCC = 0.82), 97% for phosphorus and sulfur sources (ACC = 100%; MCC = 1.0 and ACC = 92.3%; MCC = 0.82, respectively) (Fig. 6). Our model's performance was notably increased by 40% post-comprehensive curation compared to the initial draft model. Our refinement reduced false positive predictions by 17, leaving only 3 unresolved mismatches. The most remarkable improvement was in nitrogen and phosphorus sources predictions. The high predictive accuracy indicates that core metabolic pathways and multiple catabolic routes of DSM20746 have been accurately reconstructed within *i*RM23NL. Consequently, the network can predict the catabolism of numerous common compounds, such as sugars and amino acids.

## Gene essentiality predictions using *i*RM23NL

Given the increased percentage of gene-associated reactions (Fig. 2, Panel C) and the high predictive accuracy of the metabolic reconstruction, we employed *i*RM23NL further to predict exploitable single gene knockouts. For this purpose, we systematically removed each biochemical reaction from the network and optimized *i*RM23NL to produce biomass using FBA. To mitigate the inherent variability of the optimization algorithms, we repeated our FBA simulation 100 times. Additionally, we employed parsimonious enzyme usage flux balance analysis (pFBA), which involves solving two sequential linear optimization problems to determine the flux distribution of the optimal solution while minimizing the total sum of flux. Then, we compared the predicted growth rates before and after introducing the simulated gene deletion. The $FC_{gr}$ between the knocked-out and wild-type growth rates was employed as a proxy for the gene's essentiality. We proceeded with condition-specific *in silico* single gene deletions. For this purpose, we utilized a minimal and a nutrient-rich medium (M9 supplemented with glucose and LB) as well as two growth media that mimic the intra-human nasal passages and the lungs of CF patients [SNM (39) and SCFM (38)]. Generally, when subjected to nutrient-limited conditions, the model predicted a higher number of genes as essential for growth, while the count of essential genes remained consistent among oxic and anoxic conditions (Fig. 7, Panel A). In total, 4 metabolic genes exhibited a partially essential effect across all tested media. This indicates that these genes promote cellular fitness, and their deletion partially impairs the bacterium's capacity to generate biomass. These genes are the Trka family potassium uptake protein (WP_005506372.1), ribulose-phosphate 3-epimerase (WP_005507411.1), glucose-6-phosphate isomerase (WP_005508482.1), and transaldolase (WP_005509117.1). The majority of essential genes are involved in nucleotide metabolism, peptidoglycan biosynthesis, or the energy metabolism. These over-represented subsystems among the identified essential genes suggest their importance in supporting the bacterium's respiration (Fig. S4). Nevertheless, in nutrient-poor conditions (M9 medium) genes from the biosynthesis of leucine (2-isopropylmalate synthase; WP_005508679.1 and 3-isopropylmalate dehydratase; WP_005507445.1), valine (ketol-acid reductoisomerase; WP_005508646.1 and dihydroxy-acid dehydratase; WP_005509229.1), and chorismate (shikimate kinase; WP_005508729.1 and 3-dehydroquinate dehydratase; WP_005504658.1) were found to be critical for the organism's survival. Tables S4 and S5 list in detail the predicted essential genes, each corresponding to specific approaches employed in this study.

Subsequently, we conducted a protein sequence homology analysis with BLAST (42) against the human proteome to identify potential targets that could be exploited in future therapeutic strategies. For this, only genes highlighted as essential in both laboratory and synthetically defined media were considered (Fig. 7, Panel B). Overall, 35 essential genes were common in LB and M9, of which 20 common genes reported homologous counterparts in the human genome. Further analysis revealed that among these genes, five genes exhibited over 50% sequence similarity with homologous

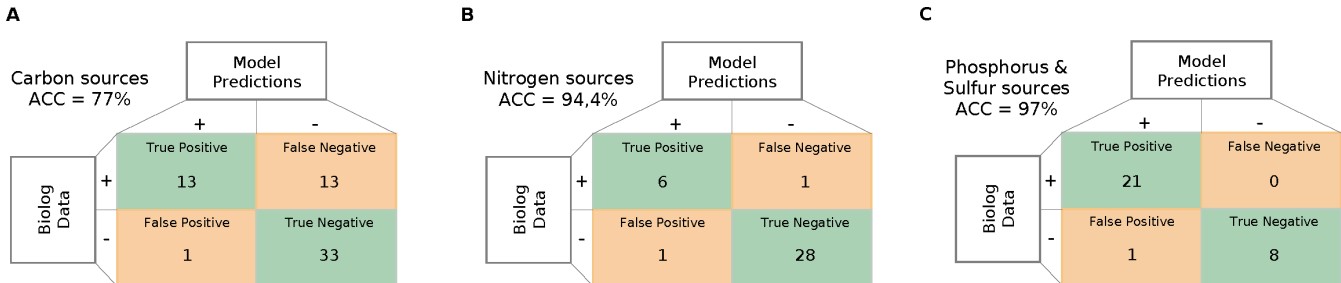

**FIG 6** Predictive accuracy performance of *i*RM23NL using nutrient utilization data. Only substrates that exhibited complete mapping to both BiGG and model identifiers could be analyzed. Green represents correct predictions, and orange represents inconsistent predictions. The overall prediction accuracy of *i*RM23NL was computed using Equation 6.

proteins although none resulted in over 80% similarity. Similarly, when *i*RM23NL was simulated with SCFM and SNM in both aerobic and anaerobic conditions, 45 shared genes were predicted to be essential. Homology analysis against the human genome yielded 25 genes with exhibited homology in the human genomes, with 7 demonstrating over 50% sequence similarity. For instance, genes encoding proteins such as phosphopyruvate hydratase (WP_005506838.1), CTP synthase (WP_044141843.1), and adenylosuccinate synthase (WP_005509175.1) consistently exhibited human counterparts with similarity exceeding 50% across all tested growth media and oxygen levels. Among the essential genes shared between both LB and M9, 15 of them did not have any homologous hits. The same was observed for 20 common essential genes

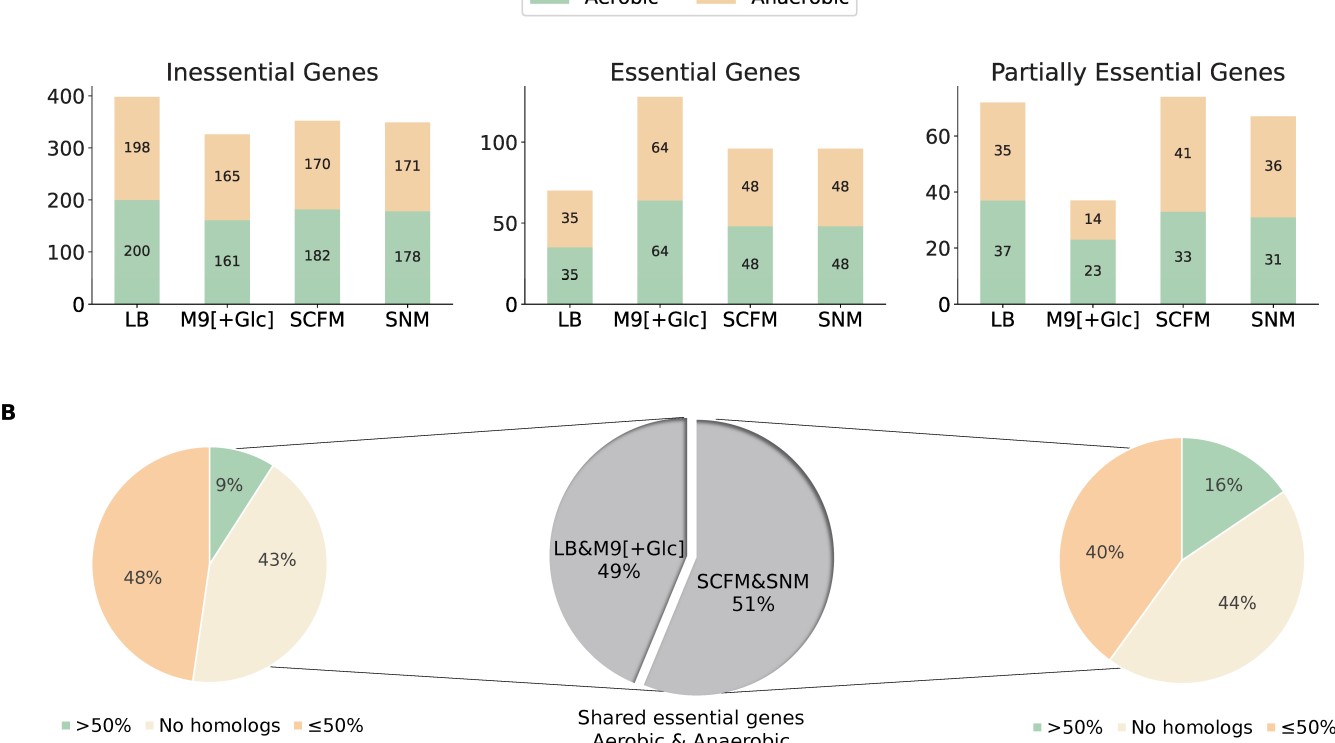

**FIG 7** Comparative analysis of novel gene essentialities in *i*RM23NL across four distinct growth media. (A) Classification of network-derived single gene deletions within *i*RM23NL, classified into essential, inessential, and partially essential genes, when subjected to aerobic (green) and anaerobic (orange) environments. Details regarding the classification schema can be found in Materials and Methods. (B) Protein sequence homology analysis of genes predicted to be essential in the laboratory media (LB and M9 pure supplemented with glucose) and the synthetically defined SNM and SCFM in both oxygen-rich and oxygen-limited conditions. The percentage identity threshold was set to 50% similarity to the human proteome.

in SCFM and SNM. Some examples of these genes include orotate phosphoribosyl-transferase (WP_005507935.1), type I pantothenate kinase (WP_005505041.1), dihydro-neopterin aldolase (WP_005507619.1), and pantetheine-phosphate adenylyltransferase (WP_005508106.1). A more detailed comparison can be found in Table S6.

Our *in silico* transposon mutant analysis using *i*RM23NL could serve as a basis for several research and practical applications from rational and condition-specific drug target development to biotechnological applications and metabolic engineering.

## DISCUSSION

The metabolic phenome of *R. mucilaginosa*, a bacterium with both beneficial and pathogenic behavior, remains still largely unexplored. Investigating its metabolic traits is of great importance as it holds the potential to unveil unique capabilities, including substrate utilization, byproduct production, and contributions to host-microbe interactions. *R. mucilaginosa* is a versatile microbe found in humans' oral, respiratory, and skin flora, where it coexists harmoniously. However, in immunocompromised individuals, *R. mucilaginosa* can act as an opportunistic pathogen, causing severe infections. Our study focuses on the metabolic aspects of *R. mucilaginosa*, particularly its behavior in isolated cultures. In 2019, a 17-species bacterial community model was reconstructed to simulate the polymicrobial community of the CF airways (43). This model accurately predicted the abundance of specific bacteria within patients' CF lung communities by linking metabolomics and 16S rRNA gene sequencing data. However, studying a bacterium's metabolism and genotype-phenotype relationships in monoculture provides a more controlled knowledge base. This allows for the precise manipulation of variables, enhancing our understanding of its individual traits, genetic makeup, metabolic pathways, and responses to stimuli (22, 23). Moreover, one can elucidate the bacterium's unique contributions to nutrient uptake, substrate production, and growth dynamics, crucial for understanding its role in a broader ecosystem. Monoculture studies identify key genes and pathways, revealing how the bacterium functions autonomously. Such analysis serves as a valuable reference, differentiating inherent characteristics from those influenced by external interactions. To this end, we empirically analyzed the metabolic phenome of *R. mucilaginosa* DSM20746 and developed the first high-quality strain-specific GEM of *R. mucilaginosa*, called *i*RM23NL. We considered literature and database organism-specific evidence to manually gap-fill the model and include highly relevant biochemical reactions. Phylogenetic analysis of further *Rothia* species provided insights into the relationship and genetic diversity between these species and was utilized to extend the metabolic network's completeness. Our model is simulation-ready, follows strictly community standards (25), and exhibits a high content quality MEMOTE score.

*R. mucilaginosa* is primarily aerobic and can perform oxic respiration by efficiently generating energy in the form of adenosine triphosphate (ATP) (1). However, when oxygen is limited or absent, *R. mucilaginosa* switches to anaerobic metabolism, which may involve fermentation or other alternative pathways to generate energy. As already mentioned, *R. mucilaginosa* has been previously found to be metabolically active in CF lungs where the oxygen levels are notably restricted (16). This indicates that the bacterium undergoes metabolic shift and can survive in microaerophilic environments. Various ROS products emerge as byproducts in the bacterial response to the fluctuating oxygen levels (34). In more detail, the cascade of ROS is initiated by the formation of $O_2^-$ upon univalent oxygen reduction within the electron transport chain (ETC). Extreme oxygen fluctuations may be lethal and can ultimately damage cellular structure. The detoxifying pathway includes the enzymes superoxide dismutase (SOD), catalase, and peroxidase that break down lethal radicals to water and oxygen enabling the cell to neutralize the oxidative stress (44) (see Fig. 4). However, the exact anaerobic respiration mechanism of *R. mucilaginosa* must be thoroughly examined in experimental settings.

Since *R. mucilaginosa*'s metabolic behavior and adaptability are mainly yet unknown, we started by testing its growth behavior in various nutrient media. Exploring how bacteria react to various growth conditions within the human body is pivotal

for understanding diseases and developing effective treatments. Moreover, they are essential for evaluating their evolution and adaptation to different environmental conditions, leading to new ecological niches in which the bacterium could be meta-bolically active. We ultimately validated *i*RM23NL using our growth kinetics data in various growth media. Overall, *i*RM23NL's predictions were in line with the experimental observations. *R. mucilaginosa* demonstrated higher experimental growth in nutrient-rich media. The model successfully simulated growth for most media, while no biomass production was achieved in the M9 pure medium. When comparing LB to RPMI, the simulated growth rate was higher in LB, while the empirical growth in RPMI was twice as high as that in LB. This can be attributed to the fact that computer models cannot mimic the entire experimental settings and lack kinetic parameters. As of September 2023, bacteria like *S. aureus*, *B. subtilis*, and *E. coli* have been extensively researched for decades, with hundreds of thousands of PubMed (45) entries since the early 1990s. In contrast, *R. mucilaginosa*'s scientific prominence only began in the 21st century, with only 423 publications to date, indicating significant knowledge gaps crucial for metabolic reconstructions. More specialized BOF would enhance the predictive power and would reflect a more organism-specific metabolism. Therefore, this scarcity underscores the urgent need for further research efforts to uncover the hidden facets of *R. mucilaginosa*'s metabolism and its significance. Notably, to simulate *in silico* growth in RPMI and SCFM media, six metal ions needed to be supplemented. These metals have also been confirmed as essential for the *in silico* growth of *S. aureus* in RPMI (41). According to the model's predictions RPMI, supplementation with manganese, zinc, and molybdate was required. Transition metals could be highly toxic; however, in controlled levels are important in the survival of all living organisms (46). For instance, they are involved in redox catalysis, needed for energy production through respiration, and in non-redox catalysis, necessary for many biosynthetic and metabolic processes. Additionally, transition metals are required for the activity of many enzymes, including those involved in genomic replication and repair and nitrogen fixation. However, since these compounds were absent from the providers' medium formulation for RPMI, we speculate that the provided medium definition may not be exact. In all cases, the suggested metal co-factor promiscuity in *R. mucilaginosa* by *i*RM23NL needs to be examined to shed light on whether the bacterium could survive in the absence of one of the suggested metals.

Moreover, we experimentally characterized the strain's ability to assimilate and utilize substrates using high-throughput phenotypic microarray assays. The utilization of various nitrogen sources did not result in active respiration, indicating that the bacterial genome lacks genes encoding for respective transporters. We used the phenotypic results to validate and extend our metabolic reconstruction, *i*RM23NL. Inconsistencies between the model and the phenotypic microarray results served as a basis for further model refinement. We enriched the model with missing transport reactions and their respective GPRs by referring to the organism- and strain-specific BioCyc (47) registry and the General Feature Format (GFF) annotation file. All in all, characterizing and determining the repertoire of nutrient sources a strain can use or assimilate is a critical factor of pathogenesis. It provides valuable insights into how pathogens adapt to host environments and evade host defenses. Our transporter-augmented model reflects a high accuracy degree with the experimental data regarding using carbon, nitrogen, phosphorus, and sulfur sources. Discrepancies between computational and empirical results highlight areas of current uncertainty knowledge regarding the metabolism of *R. mucilaginosa*. They could be attributed to non-metabolic factors that fall beyond the metabolic models' scope, including regulatory processes, gene expression, and signaling pathways. However, targeted experiments are needed to fill the remaining network gaps and reveal novel enzymatic processes.

Considering the predictive precision of our metabolic reconstruction, we utilized *i*RM23NL to derive novel hypotheses. We examined the effects of condition-specific single gene knockouts on the bacterial capacity to produce biomass. Gene essentiality

analysis is inherently contingent upon specific conditions. In the context of constraint-based metabolic modeling, a plethora of constraints are established, with the availability of nutrients, often the growth medium, being the most prevalent. By altering the availability of these nutrients, the environmental conditions are modified, consequently exerting a profound influence on the metabolic state and growth of an organism (48, 49). However, the true strength and versatility of GEMs lie in their ability to rapidly generate condition-specific hypotheses on a large scale, circumventing the need for labor-intensive and expensive screenings that may not always yield direct success. Various models, spanning organisms like *A. baumannii*, *E. coli*, *S. cerevisiae*, *P. falciparum*, and *P. aeruginosa*, demonstrated predictive accuracies ranging from 72% to 93% (50–60). Additionally, gene essentiality analysis has been instrumental in identifying potential drug targets for diseases such as cancer and viral infections, aligning well with both *in vitro* (61–63) and *in vivo* (64) data. Therefore, we utlized our GEM and created a high-throughput *in silico*-derived transposon mutant library considering two standard growth media, LB and M9, along with two growth media formulated to mimic the environment within the human body, SNM and SCFM. In this regard, we identified putative essential and partially essential genes and assessed their potential vulnerability under varying nutritional environments. With this, we opted for detecting candidate genes that could be considered in future antimicrobial and anti-inflammatory strategies in immunocompromised and CF patients. With this, we opted for identifying candidate genes for future research that hold promise for experimental validation. Determining which essential genes have human counterparts is of great importance for antibiotic drug development, as it helps assess potential side effects and cross-species effects on human genes targeted by antibiotics. Moreover, it provides insights into the molecular mechanisms of host-pathogen interactions, explaining how pathogens manipulate host cells and evade the immune system. Utilizing our GEM offers promising venues for future targeted engineering strategies without the need for laborious large-scale screening of knockouts and mutant libraries. This methodology would facilitate the rapid design of metabolic gene knockout strains by eliminating the associated reaction(s) from the model. Finally, CF lungs represent a highly dynamic environment (65, 66). However, GEMs are adaptable and can be tailored to reflect the metabolic capabilities of bacteria across diverse environmental conditions.

The main objective in our endeavor to combat *R. mucilaginosa* as an opportunistic pathogen causing infections (7) is identifying essential genes, particularly those without human counterparts. Determining these essential genes is crucial as we aim to neutralize the pathogen without harming the host. Simultaneously, we are exploring *R. mucilaginosa* as an agent with anti-inflammatory properties (18). In this context, we opt for promoting *Rothia*'s growth, focusing on modulating the environmental conditions that have been reported to do so. Once the key pathways involved in the beneficial functions of *R. mucilaginosa* are known, our gene essentiality predictions can be exploited to boost activation of these pathways. Nonetheless, being aware of *R. mucilaginosa*-specific essential genes is crucial to avoid inadvertently targeting them during therapeutic treatments, ensuring both the bacterium's growth and its anti-inflammatory activities. With this dual perspective, we indicate *R. mucilaginosa*'s therapeutic variety, including developing strategies to combat the bacterium, when it is detrimental while increasing cell biomass production when its anti-inflammatory properties are beneficial. The latter could benefit human health in the context of cystic fibrosis. However, these model-driven hypotheses need to be extensively validated via *in vitro* and *in vivo* experiments.

Altogether, creating a genome-scale metabolic network for *R. mucilaginosa* reveals insights that would have been resource-intensive to acquire using traditional wet-lab means. Understanding the metabolic complexities of *R. mucilaginosa* is essential for expanding our basic understanding of bacterium's microbiology and would benefit various practical applications. In medicine, it could facilitate the development of strategies to deal with caused infections, while in biotechnology, it would allow us to use its metabolic abilities for bioprocessing and bioengineering purposes. Hence,

our high-quality metabolic network, *i*RM23NL, could provide a systematic and detailed framework for analyzing *R. mucilaginosa*'s metabolism, yielding valuable insights with broad-reaching impacts.

## MATERIALS AND METHODS

### Experimental settings

#### *Bacterial strain and growth conditions*

The *R. mucilaginosa* DSM20746 (ATCC 25296) used for the experimental work in this study is a type strain, and it was purchased from the American Type Culture Collection (ATCC, US). To create an inoculum, the bacterium was streaked onto nutrient agar (NA, Neogen, Heywood, UK) plates from a cryopreserved glycerol stock stored at −80°C using a sterile loop. Subsequently, the plates were incubated at 37°C for 48 h to form colonies (pure cultures). It is important to note that each biological replicate was conducted using pure cultures derived from the initial frozen stock (no sub-culturing). This ensures maintaining the genetic and phenotypic characteristics of the strain without introducing any potential mutations or adaptations.

#### *Growth kinetics protocol*

*R. mucilaginosa* overnight liquid cultures were prepared by adding bacterial colonies from pure cultures to 5 mL BHI (Neogen, Heywood, UK) and were put at 37 mL in a shaking incubator for 24 h. The initial OD was assessed and, if necessary, adjusted via up-concentration or dilution to achieve $OD_{590nm} = 0.25$. Then, the bacterial suspension was subjected to centrifugation at 10,000 RPM for 5 min, and the resulting pellet was re-suspended in the medium of interest at a dilution of 1:10. Ultimately, the inoculated growth media were transferred to a sterile 96 well-plate, including three technical replicates for each tested condition together with their corresponding control conditions (sterile growth media). The outer wells were filled with milliQ water (MQ) to prevent evaporation. The respective $OD_{590nm}$ was measured aerobically at three distinct time points (0, 24, and 48 h) using an EnVision microplate reader (Perkin Elmer, Waltham, MA, USA). The microplates were incubated at 37°C during the interim periods between measurements. The final growth curves were generated for three biological replicates ($n = 3$) for the following growth media: BHI (baseline medium), LB (Neogen, Heywood, UK), M9 pure, RPMI medium (RPMI-1640 Sigma-Aldrich), and TSB (Neogen, Heywood, UK). In the M9 pure medium, only salts were considered. For detailed information regarding the constitution of M9, see Table S1. The rest of the media were prepared according to the providers' instructions.

The raw data were normalized by subtracting the blank values from the measured ODs and were summarized by calculating the arithmetic mean across all replicates. To interpret the growth of bacterial cells in all tested media and compare their growth characteristics, we employed the $FC_{OD}$ ratio, which is defined as follows:

$$FC_{OD} = \frac{OD_{590nm}^{t\ =\ 48h}}{OD_{590nm}^{t\ =\ 0h}} \tag{1}$$

In this context, we define $FC_{OD}$ below 1.4 as no growth, while $FC_{OD}$ ratios greater than 1.4 indicate a growth increase over time. This $FC_{OD}$ threshold was chosen by analyzing experimental data and growth curves. A value of 1.4 was selected, considering the range of calculated growth rates. Statistical tests, as described below, validated the threshold's reliability in accurately discerning growth from no growth in the bacterial cultures.

## Phenotypic microarray screenings

DSM20746 was tested for utilizing multiple carbon, nitrogen, phosphorus, and sulfur sources. Biolog Phenotype Microarrays (PM, Hayward, CA, USA) were employed to test the utilization of 190 carbon (PM1 and PM2A), 95 nitrogen (PM3B), 59 phosphorus (PM4A), and 35 sulfur sources (PM4A). These assays use a tetrazolium redox dye to enable a colorimetric detection of active cell respiration across different nutrient sources (40). Normal cell respiration is indicated by the formation of a purple color as a result of the reduction of the colorless dye during incubation.

The PM plates were prepared following the manufacturer's protocol for Gram-positive bacteria. Table 1 lists the assay set up for of PM plates. However, modifications were made during the cell suspension preparation. The strain was grown on nutrient agar plates without undergoing sub-culturing. Using an inoculation loop, individual colonies were picked and suspended in an inoculating fluid (IF-0) at an absorbance of 0.0915 at 590 nm. Per the established protocol, 81% of transmittance (T) should be achieved. Given our measurement of OD, the subsequent conversion of transmittance to absorbance was carried out employing the following formula:

$$\text{Absorbance} = 2 - \log_{10}(\%T) \tag{2}$$

In each well of a 96 well-plate, we introduced 100 µL of cell suspension, followed by a 48 h incubation period at 37°C. Bacterial growth was measured by the OD at 590 nm using a VICTOR Nivo Multimode microplate reader. Each PM plate was tested in duplicate.

The subsequent analysis of the acquired data included calculating the arithmetic mean across all technical and biological replicates for all measured $n$ time points. Background noise was also removed, and the data were normalized by subtracting the blank values from the actual measurements. The area under curve (AUC) was used to distinguish between growth (AUC ≥ 50) and no growth (AUC < 50). The computation of the AUCs was carried out by leveraging the linear trapezoidal rule that expresses the interpolation between data points:

$$\text{AUC}_{(t_{i+1} - t_i)} = \int_{t_i}^{t_{i+1}} f(x)\, dx \approx (t_{i+1} - t_i) \cdot \frac{1}{2}(\text{OD}_{t_{i+1}} + \text{OD}_{t_i}) \tag{3}$$

where $t_i$ is the respective measured time point and $i \in \{0, ..., e\}$. More specifically, the trapezoidal rule is iteratively applied to adjacent data points defined along the curve whose summation resulted in the final AUC value. Hence for $n$ measured data points, the final AUC value is defined as follows:

$$\text{AUC}_{t_e} = \sum_{i=0}^{e-1} \text{AUC}_{(t_{i+1} - t_i)} \tag{4}$$

Finally, we repeated this across the spectrum of tested compounds within the microarray plates.

**TABLE 1** Assay configuration for diverse Biolog PM microplates combinations[a]

|  | For 1× PM |
|---|---|
| IF-0a GN/GP (1.2×) | 10.0 |
| Dye mix (100×) | 0.12 |
| PM additive (12x) | 1.0 |
| 81%T cell suspension | 0.88 |
| Total volume | 12.0 |

[a]Volumes are expressed in µL. The provided volume quantities are adequate for inoculating the specified number of plates in this study, using 100 µL/well with an additional excess.

## Independent confirmatory testings of Biolog data

To independently confirm the Biolog data, we applied the growth kinetics protocol described above to 10 compounds. Although the base inoculating fluid (IF) used for the metabolic PM plates is proprietary, it is considered to reflect a minimal medium composed mainly of salts and buffers (40, 67). Hence, we used the M9 pure medium supplemented with different substrates to perform the independent tests (Fig. S2). The following compounds were examined: α-D-glucose, D-mannose, adonitol, L-ornithine, L-methionine, salicin, succinate, L-alanine, L-malate, and L-histidine. We also included negative controls of substrates with the Biolog inoculation fluid zero (IF-0). To ensure accuracy, triplicates were carried out for each tested compound. The M9 pure medium and the exact concentrations of added substrates are described in Table S1. All bacterial cell suspensions were prepared in 1:10 dilutions, and the $ODs_{590nm}$ were measured at 0, 24, and 48 h using an EnVision microplate reader (Perkin Elmer, Waltham, MA, US) and the associated software package.

We computed the arithmetic mean across the three replicates from the collected data set for each measured time point. Additionally, we performed a background correction to mitigate the influence of background noise or unwanted signal interference present in the measured ODs.

## Statistical hypothesis analysis

We conducted statistical tests to evaluate the chosen threshold and potential statistically significant differences between measurements at the initial and final time-points, thereby indicating the significant growth or no growth. Specifically, we employed the Student's *t*-test for each experimental condition, taking into account the data from the three biological replicates. The null hypothesis is formulated as following: there is no significant difference between the measured OD values in starting and end time-points. Prior to hypothesis testing, we checked the correctness of associated assumptions. More specifically, we assessed data normality through the Shapiro–Wilk test and verified the homogeneity of variances using the Levene's test.

## Computational framework and modeling methodology

### Phylogenomic analysis

We supported the gap-filling process using evidence of closely related species within the *Rothia* genus. Employing ANIclustermap v.1.1.0 (68), we conducted a comprehensive genomic comparison involving *R. mucilaginosa* DSM20746 and 12 distinct *Rothia* species: *R. koreensis*, *R. kristinae*, *R. santali*, *R. halotolerans*, *R. aeria*, *R. dentocariosa*, *R. terrae*, *R. amarae*, *R. nasimurium*, *R. mucilaginosa*, *R. aerolata*, *R. nasisuis*, and *R. endophytica* (see Fig. 3). In brief, ANIclustermap creates an all-vs-all genome ANI clustermap and groups microbial genomes based on their genetic similarity. ANI is a pairwise measure to classify bacterial genomes according to their genetic similarity. It is defined as the genetic similarity across all orthologous genes shared between any two genomes (69, 70). It serves as a powerful tool for distinguishing strains of the same species or closely related species.

### Draft model reconstruction and curation

The proteome of *R. mucilaginosa* DSM20746 (GCF_000175615.1) served as the basis for reconstructing a draft metabolic network. The DSM20746 (ATCC 25296) represents a type strain obtained from the throat, and its genetic and proteomic sequences were retrieved from National Centre for Biotechnology Information (NCBI) (https://www.ncbi.nlm.nih.gov). The genome sequence was annotated using the NCBI Prokaryotic Genome Annotation Pipeline (PGAP) (71). An initial draft model was built using CarveMe 1.5.1 (27). CarveMe uses mixed-integer linear programming (MILP) to convert a universal model into an organism-specific one by deleting metabolites and reactions

with low occurrence scores within the specific organism of interest. The universal BOF might yield incorrect gene essentiality predictions for biosynthesis pathways that rely on precursors unique to Gram-positive bacteria due to the absence of specific membrane and cell wall information. Hence, we chose the specialized Gram-positive template instead of the universal one to build our model more accurately. This Gram-positive template incorporates cell wall and membrane components specific to Gram-positive bacteria in contrast to the universal biomass reaction defined in CarveMe (27). Specifically, the BOF developed for *i*RM23NL includes essential macromolecules such as nucleotides and amino acids, co-enzymes, and inorganic ions. Moreover, it encompasses cell wall components like lipoteichoic acids, a peptidoglycan unit, and glycerol teichoic acids. The growth-associated (GAM) energy requirements are integrated into the biomass reaction (labeled as Growth), while non-growth associated maintenance (NGAM) is explicitly considered in the model, expressed by the reaction ATPM.

We conducted an extensive two-staged iterative gap-filling to address incomplete or missing information in the metabolic model. Gaps or missing reactions can arise for various reasons, such as incomplete genome annotations or undiscovered enzymatic activities. For this purpose, we leveraged information from both the bibliome and biochemical databases, including BioCyc (47). Thus, we ensured that the model could support the growth and viability of the organism under specific conditions.

We applied our previously published pipeline (26) to curate further the model based on community standards. The pipeline consists of eight steps, from which five (step 3–step 4) are related to model curation and ensure a high quality of the final model. In Summary, ModelPolisher (34) and SBOannotator (35) were employed to enrich the model with multiple cross-references, while the mass- and charge-unbalanced reactions were fixed. Further annotations integrated into the model encompassed: Evidence and Conclusion Ontology (ECO) terms representing the confidence level and the assertion method (biological qualifier: BQB_IS_DESCRIBED_BY), KEGG (30) subsystems as *groups:member* (biological qualifier: BQB_OCCURS_IN), and gene annotations. The latter was done by mapping the gene locus tags to the old tags using the GenBank GFF (72). Finally, we checked the presence of potential EGCs that could bias the final predictions (73). To manipulate the model structure, we employed the libSBML library (74).

The SBML Validator from libSBML (74) was used to assure a correct syntax of the model, while the quality control was carried out employing MEMOTE (36). However, it is worth noting that, as we discussed in our previous publication, MEMOTE considers only the parent nodes of the SBO directed acyclic graph excluding their respective children (26). Hence, MEMOTE was used carefully and not as an absolute quality indicator.

### *Linear programming: formulation of assumptions and constraints*

FBA is used to determine the flux distribution through optimization of the objective function, typically the maximization of biomass production rate, under steady-state conditions (21). To address the under-determined nature of the system, constraints are imposed to define an allowable solution space that aligns with cellular functions. These constraints, encompassing mass balance, thermodynamics, and capacity, contribute to the FBA maximization problem. The linear programming problem used to obtain growth rates is described as follows:

$$
\begin{aligned}
\text{maximize} \quad & Z = c^T \vec{v} \\
\text{subject to:} \quad & \mathbf{S} \cdot \vec{v} = 0 \\
& v_{\min} \leq v_r \leq v_{\max} \qquad \text{for } r \in \{1, \ldots, n\} \\
& \forall r \in I : 0 \leq v_r
\end{aligned}
\tag{5}
$$

where $\vec{v}$ is the vector of fluxes within the network, $\mathbf{S}$ is the stoichiometric matrix, $Z$ is the linear objective function, $\vec{c}$ is the vector of coefficients, and $I$ represents an index set containing the indices of all irreversible reactions. The dimensionality of vector $\vec{v}$

matches the number of reactions, denoted as $n$ in the system, and is consistent with the $n$ columns in the matrix **S**.

The unit mmol/($g_{DW} \cdot$ h) is utilized to denote the predicted growth rates since the biomass consistency (rate at which biomass is produced per unit of dry weight in the model) of *i*RM23NL approaches 1 mmol/($g_{DW} \cdot$ h) (36, 75, 76). Consequently, direct comparisons can be made between experimentally observed and predicted growth rates. These conversions maintain their validity under the assumptions of constant volume and steady-state inherent in constraint-based modeling (75).

### Bacterial growth analysis and nutrient utilization assays

Bacterial cell growth within various media and multiple substrate utilization evaluations were determined by solving Equation 5. The medium and the nutrient source of interest defined additional constraints. To achieve this objective, adjustments were made to the upper and lower limits of exchange reactions, as appropriate. We set specific uptake rates for key components within the growth medium as follows: the uptake rate of transition metals was set at 5.0 mmol/($g_{DW} \cdot$ h), the uptake rate of oxygen under aerobic conditions was established at 20.0 mmol/($g_{DW} \cdot$ h), and the rest media components equal to 10.0 mmol/($g_{DW} \cdot$ h). As previously mentioned, the M9 pure medium was used for the substrate utilization assays. Only substrates present in the metabolic network as intra- or extracellular metabolites were considered for the *in silico* validation. The results from the experimental and the *in silico* growth tests were categorized into "growth" (G) or "non-growth" (NG). Here, "growth" indicates the network's ability to generate biomass and, therefore, a positive growth rate. The model's overall prediction performance was assessed using the following statistical parameters: overall agreement (ACC):

$$ACC = \frac{TP + TN}{TP + TN + FP + FN} \tag{6}$$

and Matthews Correlation Coefficient (MCC):

$$MCC = \frac{(TP \cdot TN - FP \cdot FN)}{\sqrt{(TP + FP)(TP + FN)(TN + FP)(TN + FN)}} \tag{7}$$

where true negative (TN) and true positive (TP) represent accurate predictions, and false negative (FN) and false positive (FP) indicate incorrect predictions. Inconsistencies were resolved via iterative manual network gap-filling. For all FBA simulations, we employed the Constraints-Based Reconstruction and Analysis for Python (COBRApy) (77) package. All growth media definitions are listed in Table S2.

### Gene lethality analysis

The *in silico* single-gene knockouts were performed as described in our previous study using FBA (26). To address the degeneracy issue of optimization, we additionally ran our FBA simulations in a total of 100 independent runs. Furthermore, we utilized pFBA, a method that allows us to ascertain the flux distribution of the optimal solution while concurrently minimizing the overall flux sum (78). The results were categorized as either essential $FC_{gr} = 0$, inessential ($FC_{gr} = 1$), or partially essential ($0 < FC_{gr} < 1$), where $FC_{gr}$ denotes the FC bacterial growth rate before and after deletion (27). Shared essential genes between FBA and pFBA, as well as all tested conditions, were further aligned against the human genome using BLAST (42).

### ACKNOWLEDGMENTS

N.L. is supported by the Deutsche Forschungsgemeinschaft (DFG, German Research Foundation) under Germany's Excellence Strategy—EXC 2124-390838134, and by the Cluster of Excellence 'Controlling Microbes to Fight Infections' (CMFI). A.D. is supported by the German Center for Infection Research (DZIF, doi: 10.13039/100009139) within

the Federal Ministry of Education and Research (BMBF, German Centers for Health Research of the Federal Ministry of Education and Research), grant no. 8020708703. The authors acknowledge the support of the Open Access Publishing Fund of the University of Tübingen (https://uni-tuebingen.de/en/216529). The authors also thank Charlotte Rigauts and Anouk Van Hauwermeire for assistance with the Biolog assays.

## AUTHOR AFFILIATIONS

[1]Computational Systems Biology of Infections and Antimicrobial-Resistant Pathogens, Institute for Bioinformatics and Medical Informatics (IBMI), Eberhard Karl University of Tübingen, Tübingen, Germany

[2]Department of Computer Science, Eberhard Karl University of Tübingen, Tübingen, Germany

[3]Cluster of Excellence 'Controlling Microbes to Fight Infections', Eberhard Karl University of Tübingen, Tübingen, Germany

[4]German Center for Infection Research (DZIF), partner site Tübingen, Tübingen, Germany

[5]Quantitative Biology Center (QBiC), Eberhard Karl University of Tübingen, Tübingen, Germany

[6]Laboratory of Pharmaceutical Microbiology (LPM), Ghent University, Ghent, Belgium

[7]Data Analytics and Bioinformatics, Institute of Computer Science, Martin Luther University Halle-Wittenberg, Halle (Saale), Germany

## AUTHOR ORCIDs

Nantia Leonidou http://orcid.org/0000-0002-0248-6679
Tom Coenye http://orcid.org/0000-0002-6407-0601
Aurélie Crabbé http://orcid.org/0000-0003-3084-4418
Andreas Dräger http://orcid.org/0000-0002-1240-5553

## FUNDING

| Funder | Grant(s) | Author(s) |
|---|---|---|
| Deutsche Forschungsgemeinschaft (DFG) | EXC 2124 - 390838134 | Nantia Leonidou |
| | | Andreas Dräger |
| Deutsches Zentrum für Infektionsforschung (DZIF) | 8020708703 | Nantia Leonidou |
| | | Andreas Dräger |

## AUTHOR CONTRIBUTIONS

Nantia Leonidou, Conceptualization, Data curation, Formal analysis, Investigation, Methodology, Project administration, Software, Validation, Visualization, Writing – original draft, Writing – review and editing | Lisa Ostyn, Investigation, Supervision | Tom Coenye, Supervision, Writing – review and editing | Aurélie Crabbé, Methodology, Supervision, Writing – review and editing | Andreas Dräger, Funding acquisition, Resources, Supervision, Writing – review and editing

## DATA AVAILABILITY

Supplementary data are available along with this article. Additionally, iRM23NL is available at the BioModels Database (79) as an SBML Level 3 Version 1 (37) file wrapped in an Open Modelling EXchange format (OMEX) (80) archive file together with a metadata file (81). Access the model at https://identifiers.org/biomodels.db/MODEL2310240001.

## ADDITIONAL FILES

The following material is available online.

## Supplemental Material

**Figure S1 (Spectrum04006-23-s0001.pdf).** Experimentally-derived catabolic phenome of *R. mucilaginosa* DSM20746.

**Figure S2 (Spectrum04006-23-s0002.pdf).** Growth curves of the independent confirmatory tests for validating the Biolog PM results.

**Figure S3 (Spectrum04006-23-s0003.pdf).** Detailed comparative analysis of gene essentiality in silico predictions using iRM23NL.

**Figure S4 (Spectrum04006-23-s0004.pdf).** Distribution of essential genes in metabolic subsystems.

**Table S1 (Spectrum04006-23-s0005.xlsx).** Reconstitution of 10X M9 medium and amount of substrates supplemented.

**Table S2 (Spectrum04006-23-s0006.xlsx).** Detailed definition of growth media used for in silico simulations.

**Table S3 (Spectrum04006-23-s0007.xlsx).** Summary of high-throughput Biolog PM results.

**Table S4 (Spectrum04006-23-s0008.xlsx).** List of essential genes predicted by FBA in different nutritional media.

**Table S5 (Spectrum04006-23-s0009.xlsx).** List of essential genes predicted by pFBA in different nutritional media.

**Table S6 (Spectrum04006-23-s0010.xlsx).** Comparative analysis of predicted essential genes for *R. mucilaginosa* in different nutrient environments.

## Open Peer Review

**PEER REVIEW HISTORY (review-history.pdf).** An accounting of the reviewer comments and feedback.

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
