## [Reviewer comments · Microbiology Spectrum]

Microbiology Spectrum

Genome-scale model of *Rothia mucilaginosa* predicts gene essentialities and reveals metabolic capabilities

Nantia Leonidou, Lisa Ostyn, Tom Coenye, Aurélie Crabbé, and Andreas Dräger

Corresponding Author(s): Nantia Leonidou, Eberhard Karls Universität Tübingen

Review Timeline:

Submission Date:	November 28, 2023
Editorial Decision:	January 9, 2024
Revision Received:	February 23, 2024
Accepted:	March 20, 2024

Editor: Angela Re

Reviewer(s): Disclosure of reviewer identity is with reference to reviewer comments included in decision letter(s). The reviewers have opted to remain anonymous.

Transaction Report:

DOI: <https://doi.org/10.1128/spectrum.04006-23>

Re: Spectrum04006-23 (Genome-Scale Modeling of *Rothia mucilaginosa* Reveals Insights into Metabolic Capabilities and Therapeutic Strategies for Cystic Fibrosis)

Dear Ms. Nantia Leonidou:

Thank you for the privilege of reviewing your work. Below you will find my comments, instructions from the Spectrum editorial office, and the reviewer comments. You are warmly invited to account comprehensively for the comments received both in technical aspects and in some conclusions drawn from your analysis.

Revision Guidelines

Sincerely,
Angela Re
Editor
Microbiology Spectrum

Reviewer #1 (Comments for the Author):

In the manuscript "Genome-Scale Modeling of *Rothia mucilaginosa* Reveals Insights into Metabolic Capabilities and Therapeutic Strategies for Cystic Fibrosis" submitted for publication in Microbiology Spectrum, Leonidou et al reconstructed a genome-scale metabolic model (GEM) of *Rothia mucilaginosa*, using semi-automatic methods for draft reconstruction and further manual

curation. Model performance was assessed with growth phenotypic data under various carbon, nitrogen, phosphorus, and nitrogen sources. The model exhibits high accuracy in predicting growth under the tested sources after curation.

MAJOR COMMENTS

The modeling work is solid and the GEM for *R. mucilaginosa* provides a valuable resource for the scientific community. However, the main concerns with the results of this paper are related to the conclusions drawn from gene essentiality predictions regarding therapeutic strategies for CF (see below). Moreover, other concerns arise regarding the definition of the FCOD threshold and the definition of the biomass objective function.

In-silico gene essentiality analysis can only go this far in assessing the importance of a gene in an organism. It is critical to notice that gene essentiality can only provide a condition-specific assessment. For CF this means that i) the authors would provide an accurate assessment of the conditions *R. mucilaginosa* encounters in the lung (and in different patients) - which is not feasible, and ii) that conditions are static and never change - also not the case. Thus, the conclusions drawn from this analysis are only speculative in nature and should (if at all) be presented this way. They do not represent a therapeutic strategy and should not be presented as such (e.g. in the title). Furthermore, it is conceivable that essential genes would generally belong to nucleotide and energy metabolism. Can you provide any in vivo studies on gene essentiality and if they match the in silico predictions?

The authors base the final analysis of this work on the reported association between *R. mucilaginosa* and CF. They provide gene essentiality predictions to identify possible targets for CF as therapeutic strategies. However, this requires solid proof that *R. mucilaginosa* has a main role in the onset or worsening of CF, other than just being comparatively favored in the environment caused by CF. Correlative data from microbiome surveys are not suitable here. Are there any references supporting that reducing *R. mucilaginosa* can alleviate CF-derived symptoms? If this has been shown, add this as support for your findings.

Additionally

R. mucilaginosa is reported as an anti-inflammatory bacterium in lung diseases (<https://www.ncbi.nlm.nih.gov/pmc/articles/PMC9068977/>). So, are your targeting strategies contrary to this study? Also, as CF is a hereditary disease, would gene therapy not be a better approach than targeting a bacteria like *R. mucilaginosa*?

Moreover, the model was tested only under aerobic growth? Are there any anaerobic growth data? While the lung can be an oxic environment, large bacterial burden rapidly reduces oxygen availability and renders it anoxic.

It would also be useful if the authors can explain (in the methods) the detailed protocol they followed to define the biomass objective function? What are the components and their compositions? How was the growth-associated and non-growth-associated ATP maintenance derived?

The authors state that they use a fold-change OD threshold of 1.4 to determine a binary response variable of growth/no growth. In the legend of Figure 4, they state that they defined it based on experimental results, and in the methods section they state that the correctness of this definition was assessed through a number of statistical tests. Please provide a detailed step-by-step description of how this threshold was chosen. Can the authors provide a short analysis showing how sensitive the results of this work are to the FCOD threshold? Would a threshold of 1.3 or 1.5 severely affect the calculated accuracy of the model?

Overall, the GEM represents a valuable resource but the concerns regarding its application for CF therapy significantly reduces my support for this manuscript. Especially the interpretation of gene essentiality and its use should be removed (or experimentally validated) from the manuscript/title since this is misleading the reader into believe that gene essentiality can be used to define interventions in an environment that is highly dynamic.

MINOR COMMENTS

Line 273-276: What were the growth rates compared to experimental conditions under anaerobic conditions? Were those comparable?

Table S6 : Should be *R. mucilaginosa* and not *T. mucilaginosa*.

Line 918, "growt" seems to be a typo.

Figure 4B, the datapoints shown are mean values of triplicates, but errorbars are not shown, please include them.

Figure 5A, color bar should display units.

Reviewer #2 (Comments for the Author):

In this study, the authors developed the first genome-scale metabolic model for *R. mucilaginosa*, named iRM23NL. This model, validated through growth kinetics and phenotypic tests, accurately predicts growth and substrate use, formulates new

hypotheses, and identifies potential antimicrobial targets. Overall, the work is sound, and the manuscript is clearly presented. I have minor comments.

1. The term "anti-inflammatory" is used in Abstract and throughout the manuscript. What does this mean exactly in the context of CF? Please clarify whether this is beneficial or harmful to the pulmonary system in patients with CF.
2. Growth rate prediction using GEM (e.g., Figure 4C): Shouldn't its unit be 1/h (specific growth rate), not "mmol/(gDW · h)"?
3. The authors mention that this organism's "genotype-phenotype relationships remain largely unknown". Despite this, has there been any examination of this organism concerning specific pathways that might be relevant to treatment? If so, please include the relevant discussion where appropriate ("Formulating novel hypotheses using iRM23NL" or "Discussion").
4. Figure 4B: The asterisks (**, ****, and ns) need to be defined in the figure legend.

Martin-Luther-Universität Halle-Wittenberg, 06099 Halle (Saale)

To the editors of
ASM Microbiology Spectrum

Ihre Zeichen
Spectrum04006-23

Ihr Schreiben vom

February 19, 2024

Unsere Zeichen

Datum

Halle (Saale), February 20, 2024

Rebuttal letter with a point-by-point response to all referee comments to our manuscript

Dear Prof. Re,

With this we are resubmitting the enclosed paper entitled “Genome-scale model of *Rothia mucilaginosa* predicts gene essentialities and reveals metabolic capabilities” for consideration as an Article in *ASM Microbiology Spectrum*.

A. Associate Editor (Remarks to Author)

We thank you and the anonymous reviewers for their detailed comments and have carefully addressed all aspects of our manuscript. A point-to-point list is given below. We have revised and enriched our manuscript with all missing information based on the reviewers’ comments, while we indicated clearly within our responses when the required information was already included in the text. Additionally, we noticed inaccuracies in the numerical values within the model statistics, (between lines 142 and 150), which we corrected. To simplify the review process, we include a version of the manuscript in which ~~deleted text is crossed out~~, **additions appear in blue**, and **changes are shown in red**. Hovering the mouse over red text in the manuscript brings up a tool-tip box with the previous version. Our replies to the reviewers below are also highlighted in **blue**. Please do not hesitate to contact us with any questions you may have regarding this manuscript.

B. Reviewer #1

In the manuscript “Genome-Scale Modeling of Rothia mucilaginosa Reveals Insights into Metabolic Capabilities and Therapeutic Strategies for Cystic Fibrosis” submitted for publication in Microbiology Spectrum, Leonidou et al reconstructed a genome-scale metabolic model (GEM) of Rothia mucilaginosa, using semi-automatic methods for draft reconstruction and further manual curation. Model performance was assessed with growth phenotypic data under various carbon, nitrogen, phosphorus, and nitrogen sources. The model exhibits high accuracy in predicting growth under the tested sources after curation.

B.1. Major comments

*The modeling work is solid and the GEM for *R. mucilaginosa* provides a valuable resource for the scientific community. However, the main concerns with the results of this paper are related to the conclusions drawn from gene essentiality predictions regarding therapeutic strategies for CF (see below). Moreover, other concerns arise regarding the definition of the FCOD threshold and the definition of the biomass objective function.*

*In-silico gene essentiality analysis can only go this far in assessing the importance of a gene in an organism. It is critical to notice that gene essentiality can only provide a condition-specific assessment. For CF this means that i) the authors would provide an accurate assessment of the conditions *R. mucilaginosa* encounters in the lung (and in different patients) - which is not feasible, and ii) that conditions are static and never change - also not the case. Thus, the conclusions drawn from this analysis are only speculative in nature and should (if at all) be presented this way. They do not represent a therapeutic strategy and should not be presented as such (e.g., in the title).*

We absolutely agree that gene essentiality analysis, analogous to other simulations done using GEMs, is condition-specific. In constraint-based metabolic models, various constraints are defined, with the most frequent one being the nutrients' availability (growth medium). Modifying the availability of nutrients leads to changes in environmental conditions, subsequently exerting a profound influence on an organism's metabolic state and growth [1, 2]. However, the underlying power and the inherent strength of GEMs is to enable the rapid formulation of new condition-specific hypotheses on a large scale without the need for laborious and costly screenings that may not yield direct success. Consequently, to approximate the environment in a cystic fibrosis (CF) lung that the bacteria encounter within human CF lungs, we used the previously defined synthetic cystic fibrosis medium (SCFM) for our simulations [3]. Moreover, we agree that CF lungs are a highly dynamic environment. However, GEMs are versatile and can be adapted to represent the metabolic capabilities of bacteria in various environmental conditions. They allow for the simulation of metabolic fluxes and responses to changing nutrient availability, making them suitable for dynamic environments.

In response to the reviewer's comment, we added a section in the Discussion describing the condition-specificity and the adaptability power of GEMs (lines 434-439, 644-657, and 684-688). In addition, we have modified and re-formulated the title to "Genome-scale model of *Rothia mucilaginosa* predicts gene essentialities and reveals metabolic capabilities" as well as the title of the gene essentiality section in the Results section to "Gene essentiality predictions using *iRM23NL*". To underscore the dependence on nutrient availability, we have incorporated the term "condition-specific" into the manuscript (see section "Formulating novel hypotheses using *iRM23NL*" and Discussion). Nevertheless, the reviewer is correct that we are not proposing a novel therapy. Hence, following the comment of the reviewer, we have modified the text (lines 469-470 and 671-673) to highlight the predictive nature of this analysis and its potential applicability for future therapy development studies.

Furthermore, it is conceivable that essential genes would generally belong to nucleotide and energy metabolism. Can you provide any in vivo studies on gene essentiality and if they match the in silico predictions?

In our study, model validation was focused on comparing growth kinetics data obtained under various media formulations, along with high-throughput nutrient utilization data. As of today, numerous *in silico* simulations of gene deletion for various microorganisms have demonstrated successful alignment with experimental studies on gene essentiality.

Models reconstructed for *A. baumannii* strains achieved accuracies of 72 % to 88.5 %, while predictions for *E. coli* and *S. cerevisiae* ranged from 87.8 % to 93.4 % [4–9]. Stage-specific models for *P. falciparum* simulated 71.2 % of experimental data, and models for *B. subtilis* and *L. kluyveri* showed accuracies between 75 % and 94 % when compared against gene lethality data [10–12]. *In vivo* gene deletion studies for *P. aeruginosa* aligned with model simulations, achieving an 85 % accuracy, and computational predictions identified synthetic lethal gene deletion pairs in yeast, successfully validated against *in vivo* gene essentiality data [13, 14]. Gene essentiality analysis has also been applied to predict drug targets in cancer and viral infections, matching both *in vitro* [15–17] and *in vivo* [18].

Following the comment of reviewer, we have summarized this information in the discussion (lines 657–664).

The authors base the final analysis of this work on the reported association between R. mucilaginosa and CF. They provide gene essentiality predictions to identify possible targets for CF as therapeutic strategies. However, this requires solid proof that R. mucilaginosa has a main role in the onset or worsening of CF, other than just being comparatively favored in the environment caused by CF. Correlative data from microbiome surveys are not suitable here. Are there any references supporting that reducing R. mucilaginosa can alleviate CF-derived symptoms? If this has been shown, add this as support for your findings. Additionally, R. mucilaginosa is reported as an anti-inflammatory bacterium in lung diseases (<https://www.ncbi.nlm.nih.gov/pmc/articles/PMC9068977/>). So, are your targeting strategies contrary to this study? Also, as CF is a hereditary disease, would gene therapy not be a better approach than targeting a bacteria like R. mucilaginosa?

We would like to apologize for the confusion caused by our gene essentiality analysis and its potential importance to CF and antimicrobial treatments. We did not intend to imply that inhibiting *R. mucilaginosa* would diminish symptoms caused by CF. Indeed, it is not fully clear what the role of this microorganism is in the CF disease process. As we mention in our manuscript, on one hand *R. mucilaginosa* could have a beneficial role in this hereditary disease via its anti-inflammatory properties [19]. On the other hand, *R. mucilaginosa* has also been occasionally associated with infections including endocarditis and lung diseases, hence there could be a need to eradicate this microorganism [20–25]. For the latter purpose, we aimed to identify essential genes in *R. mucilaginosa*, particularly those lacking human counterparts. Furthermore, from the perspective that *R. mucilaginosa* could potentially have beneficial properties in the context of CF, our gene essentiality predictions could also have applications of interest. Specifically, the production of enzymes encoded by the predicted essential genes could be enhanced to boost the bacterium's growth and its anti-inflammatory properties. This dual perspective guides our strategy, encompassing the development of techniques to combat *R. mucilaginosa* when it poses a threat while simultaneously enhancing cell biomass production when its anti-inflammatory properties prove to be advantageous. This comprehensive approach holds potential therapeutic benefits, also connected to CF.

We have now enriched our Discussion section (lines 689–711) with information on explaining how the predicted essential genes could be useful in the CF context.

Regarding the comment of the reviewer on gene therapies for CF, we agree that directly targeting the genetic defect in this patient population is an important area of research. Yet, the recent development of cystic fibrosis transmembrane conductance regulator (CFTR) modulator therapies for patients with CF, which are correcting the function of the CFTR protein, indicate that bacterial infections in CF patients are still clinically important and that (novel) antimicrobial strategies are still needed.

Moreover, the model was tested only under aerobic growth? Are there any anaerobic growth data? While the lung can be an oxic environment, large bacterial burden rapidly reduces oxygen availability and renders it anoxic.

R. mucilaginosa is known to be a facultatively anaerobic bacterium since its first isolation [26], with clinical strains exhibiting the same trait [27, 28]. Facultative anaerobes possess the ability to adjust their metabolism, allowing them to thrive under both aerobic and anaerobic conditions, albeit with a preference for aerobic environments. For instance, the literature confirms this characteristic for a member of the *Rothia* genus from the family *Micrococcaceae*, *R. dentocariosa* [29]. Consequently, it is anticipated that growth under anaerobic conditions will be less compared to aerobic growth, aligning with the predictions of our model (see Figure 4C). Despite less optimal growth under anaerobic conditions, our group found that *R. mucilaginosa* exerts its anti-inflammatory properties both under aerobic and anaerobic growth conditions [19]. Hence, for practical reasons we chose to focus our experimental validation of the model on aerobic conditions.

It would also be useful if the authors can explain (in the methods) the detailed protocol they followed to define the biomass objective function? What are the components and their compositions? How was the growth-associated and non-growth-associated ATP maintenance derived?

Due to the lack of experimental data, particularly regarding biomass composition, for *R. mucilaginosa*, we considered obtaining a realistic biomass objective function during the CarveMe-driven draft reconstruction. To achieve this, we utilized the Gram-positive template outlined in lines 895–901 of the submitted manuscript [30]. The resulting biomass reaction incorporates essential macromolecules such as nucleotides and amino acids, co-enzymes, inorganic ions, and both growth and non-growth associated maintenance costs (e.g., ATP, ADP, and P).

Furthermore, the model encompasses essential components such as membrane and cell wall constituents, including glycerol teichoic acids, lipoteichoic acids, and a peptidoglycan unit. The growth-associated energy requirements (GAM) are embedded within the biomass reaction (labelled as Growth), while non-growth associated maintenance (NGAM) is explicitly considered in the model, expressed by the reaction ATPM. The presence of both energy-related reactions can be validated through the metabolic model test suite (MEMOTE) report under the “Energy Metabolism” section. In summary, the final biomass reaction is standardized to represent 1 g of cell dry weight.

We have now enriched our biomass-related statement within Materials and Methods with this information (lines 901–913).

The authors state that they use a fold-change OD threshold of 1.4 to determine a binary response variable of growth/no growth. In the legend of Figure 4, they state that they defined it based on experimental results, and in the methods section they state that the correctness of this definition was assessed through a number of statistical tests. Please provide a detailed step-by-step description of how this threshold was chosen. Can the authors provide a short analysis showing how sensitive the results of this work are to the FCOD threshold? Would a threshold of 1.3 or 1.5 severely affect the calculated accuracy of the model?

The FC_{OD} threshold was initially chosen by looking at the experimental data and the associated growth curves. Very high fold-changes could directly be classified as growth by only looking at the growth curves. After careful consideration of the remaining growth curves, a threshold of 1.4 was chosen by considering the ranges in which the calculated growth rates were. To ensure the robustness of the chosen threshold, its correctness was rigorously validated through statistical tests, as outlined in the legend of Figure 4, and detailed in the methods section. These tests provide additional confidence in the reliability of the selected threshold and its ability to accurately discriminate between growth and no growth in the bacterial cultures under investigation.

Following the comment of the reviewer, we included the description in the methodology (lines 776–781).

Minor alterations to the threshold within a reasonable range do not significantly impact the accuracy of the model. For instance, adjusting the threshold to 1.3 or 1.5 only affects the assessment of growth in M9 medium, transitioning from no growth to growth, while the model predicts no growth. In more detail, a threshold 1.3 would represent no growth, while 1.5 would result in no growth that aligns with the model prediction.

Overall, the GEM represents a valuable resource but the concerns regarding its application for CF therapy significantly reduces my support for this manuscript. Especially the interpretation of gene essentiality and its use should be removed (or experimentally validated) from the manuscript/title since this is misleading the reader into believe that gene essentiality can be used to define interventions in an environment that is highly dynamic.

We have carefully addressed the concerns of the reviewer regarding the applications of this work for CF and would like to refer the reviewer to our response above. Given the implementation of the suggested modifications to the manuscript and the clarifications throughout the text about the role of *R. mucilaginosa* in the CF disease process, we would kindly like to retain the section of gene essentiality predictions. We firmly believe that these computational predictions can significantly contribute to and foster the study of *R. mucilaginosa* in and outside the context of CF.

B.2. Minor comments

Line 273–276: What were the growth rates compared to experimental conditions under anaerobic conditions? Were those comparable?

As mentioned above, *R. mucilaginosa*, known as a facultative anaerobe, can adapt to both aerobic and anaerobic environments [26]. Growth under anaerobic conditions is expected to be less compared to aerobic growth, as predicted by the model as well (see Figure 4C and lines 285–289). Therefore, we focused our experimental evaluation solely on aerobic conditions. The predicted anaerobic growth rates were lower than the aerobic ones, verifying the facultative anaerobic nature of *R. mucilaginosa* strains [27, 28].

Table S6 : Should be R. mucilaginosa and not T. mucilaginosa.

The typo has been corrected.

Line 918, “growt” seems to be a typo.

The typo has been corrected.

Figure 4B, the datapoints shown are mean values of triplicates, but errorbars are not shown, please include them.

Errorbars have now been added in Figure4B, as well as in the supplementary figures.

Figure 5A, color bar should display units.

The units have been added to the color bar.

C. Reviewer #2

*In this study, the authors developed the first genome-scale metabolic model for *R. mucilaginosa*, named iRM23NL. This model, validated through growth kinetics and phenotypic tests, accurately predicts growth and substrate use, formulates new hypotheses, and identifies potential antimicrobial targets.*

Overall, the work is sound, and the manuscript is clearly presented. I have minor comments.

C.1. Minor comments

The term “anti-inflammatory” is used in Abstract and throughout the manuscript. What does this mean exactly in the context of CF? Please clarify whether this is beneficial or harmful to the pulmonary system in patients with CF.

In the context of CF, commensal bacteria with anti-inflammatory properties may have a beneficial role as they could potentially decrease the pathological inflammation in the lung environment. The anti-inflammatory properties of *R. mucilaginosa* were recently demonstrated by Rigauts *et al.*, wherein *R. mucilaginosa* inhibited pro-inflammatory responses induced by pathogens or lipopolysaccharide both *in vitro* in a three-dimensional cell culture model and *in vitro* an *in vivo* mouse model [19]. Consequently, the presence of *R. mucilaginosa* within the lower respiratory tract could potentially be beneficial for CF patients. Nevertheless, *R. mucilaginosa* has also been occasionally associated with infections including in the context of CF and other lung diseases, hence there could be a need to eradicate this microorganism. Given the latter, it remains important to develop strategies to eradicate this microorganism, when needed.

In response to the reviewer’s comment, we have enriched the discussion of our manuscript by describing the dual role of *R. mucilaginosa* in health and disease (lines 689–711).

Growth rate prediction using GEM (e.g., Figure 4C): Shouldn’t its unit be 1/h (specific growth rate), not “mmol/(g_{DW} · h)”?

The use of mmol/(g_{DW} · h) as a unit for growth rate holds true under the condition that the biomass consistency of a metabolic model approaches 1 mmol/(g_{DW} · h) [31–33]. When this consistency is close to 1 mmol/(g_{DW} · h), it allows for a direct comparison between experimentally observed growth rates, often given in the unit 1/h, and the growth rates calculated within the metabolic model. Our model reported a biomass consistency of 1.03 mmol/(g_{DW} · h) validating the correct use of this unit. Finally, these conversions hold true since constraint-based modelling assumes constant volume and steady state [34]. Nevertheless, the growth rate in biological systems is determined by the flux through the biomass objective function, indicating the system’s biomass-producing reaction. Typically expressed as mmol/(g_{DW} · h), where g_{DW} denotes grams of dry weight, this unit is standardized within the Systems Biology Markup Language (SBML) specification, ensuring consistency and compatibility across metabolic models [33, 34].

In response to the reviewer’s comment, we have incorporated this clarification into the Material and Methods section (lines 964–971).

The authors mention that this organism’s “genotype-phenotype relationships remain largely unknown”. Despite this, has there been any examination of this organism concerning specific pathways that might be relevant to treatment? If so, please include the relevant discussion where appropriate (“Formulating novel hypotheses using iRM23NL” or “Discussion”).

In the existing literature, *R. mucilaginosa* has primarily been investigated within the framework of microbial communities, focusing on understanding the metabolic mechanisms that contribute to cross-feeding interactions within these communities [35–37].

In the realm of investigating metabolic pathways for potential treatment strategies, only two scientific publications have delved into the role of *R. mucilaginosa* pathways in antimicrobial approaches. Uranga *et al.* elucidate how the enterobactin produced by *R. mucilaginosa* hinders the bacterial growth of oral microbiota, including cariogenic *S. mutans*, *A. timonensis*, and *Streptococcus sp.*, along with four methicillin-resistant *S. aureus* strains [38]. The authors suggest that this finding may pave the way for developing innovative therapeutic avenues against highly resistant pathogens. Additionally, Rigauts *et al.* unveil the potential of *R. mucilaginosa* in the lower airways to alleviate inflammation through NF- κ B pathway inactivation, thereby influencing the severity of chronic lung diseases [19].

The evidence mentioned has already been integrated into the introduction section of the submitted manuscript (lines 37–72).

*Figure 4B: The asterisks (**, ***, and ns) need to be defined in the figure legend.*

As defined in the figure caption, the asterisks indicate the significance levels, as determined with statistical testing. To avoid overloading the figure and the figure legend with information, we would keep the definition in the figure caption.

On behalf of the authors, sincerely yours,

Prof. Dr. Andreas Dräger

References

1. Camacho, A., Rochera, C. & Picazo, A. Effect of experimentally increased nutrient availability on the structure, metabolic activities, and potential microbial functions of a maritime Antarctic microbial mat. *Frontiers in Microbiology* **13**, 900158 (2022).
2. Gasperotti, A., Brameyer, S., Fabiani, F. & Jung, K. Phenotypic heterogeneity of microbial populations under nutrient limitation. *Current opinion in biotechnology* **62**, 160–167 (2020).
3. Turner, K. H., Wessel, A. K., Palmer, G. C., Murray, J. L. & Whiteley, M. Essential genome of *Pseudomonas aeruginosa* in cystic fibrosis sputum. *Proceedings of the National Academy of Sciences* **112**, 4110–4115 (2015).
4. Presta, L., Bosi, E., Mansouri, L., Dijkshoorn, L., Fani, R. & Fondi, M. Constraint-based modeling identifies new putative targets to fight colistin-resistant *A. baumannii* infections. *Scientific reports* **7**, 1–12 (2017).
5. Norsigian, C. J., Kavvas, E., Seif, Y., Palsson, B. O. & Monk, J. M. iCN718, an updated and improved genome-scale metabolic network reconstruction of *Acinetobacter baumannii* AYE. *Frontiers in genetics* **9**, 121 (2018).
6. Zhao, J., Zhu, Y., Han, J., Lin, Y.-W., Aichem, M., Wang, J., Chen, K., Velkov, T., Schreiber, F. & Li, J. Genome-scale metabolic modeling reveals metabolic alterations of multidrug-resistant *Acinetobacter Baumannii* in a murine bloodstream infection model. *Microorganisms* **8**, 1793 (2020).
7. Zhu, Y., Zhao, J., Maifiah, M. H. M., Velkov, T., Schreiber, F. & Li, J. Metabolic responses to polymyxin treatment in *Acinetobacter baumannii* ATCC 19606: integrating transcriptomics and metabolomics with genome-scale metabolic modeling. *Msystems* **4**, e00157–18 (2019).

8. Monk, J. M., Lloyd, C. J., Brunk, E., Mih, N., Sastry, A., King, Z., Takeuchi, R., Nomura, W., Zhang, Z., Mori, H., *et al.* iML1515, a knowledgebase that computes *Escherichia coli* traits. *Nature biotechnology* **35**, 904–908 (2017).
9. Joyce, A. R., Reed, J. L., White, A., Edwards, R., Osterman, A., Baba, T., Mori, H., Lesely, S. A., Palsson, B. Ø. & Agarwalla, S. Experimental and computational assessment of conditionally essential genes in *Escherichia coli*. *Journal of bacteriology* **188**, 8259–8271 (2006).
10. Abdel-Haleem, A. M., Hefzi, H., Mineta, K., Gao, X., Gojobori, T., Palsson, B. O., Lewis, N. E. & Jamshidi, N. Functional interrogation of *Plasmodium* genus metabolism identifies species- and stage-specific differences in nutrient essentiality and drug targeting. *PLoS computational biology* **14**, e1005895 (2018).
11. Oh, Y.-K., Palsson, B. O., Park, S. M., Schilling, C. H. & Mahadevan, R. Genome-scale reconstruction of metabolic network in *Bacillus subtilis* based on high-throughput phenotyping and gene essentiality data. *Journal of Biological Chemistry* **282**, 28791–28799 (2007).
12. Nanda, P., Patra, P., Das, M. & Ghosh, A. Reconstruction and analysis of genome-scale metabolic model of weak Crabtree positive yeast *Lachancea kluyveri*. *Scientific Reports* **10**, 16314 (2020).
13. Oberhardt, M. A., Puchałka, J., Fryer, K. E., Martins dos Santos, V. A. & Papin, J. A. *Genome-scale metabolic network analysis of the opportunistic pathogen Pseudomonas aeruginosa PAO1* 2008.
14. Harrison, R., Papp, B., Pál, C., Oliver, S. G. & Delneri, D. Plasticity of genetic interactions in metabolic networks of yeast. *Proceedings of the National Academy of Sciences* **104**, 2307–2312 (2007).
15. Aller, S., Scott, A., Sarkar-Tyson, M. & Soyer, O. S. Integrated human-virus metabolic stoichiometric modelling predicts host-based antiviral targets against Chikungunya, Dengue and Zika viruses. *Journal of The Royal Society Interface* **15**, 20180125 (2018).
16. Bidkhori, G., Benfeitas, R., Elmas, E., Kararoudi, M. N., Arif, M., Uhlen, M., Nielsen, J. & Mardinoglu, A. Metabolic network-based identification and prioritization of anticancer targets based on expression data in hepatocellular carcinoma. *Frontiers in physiology* **9**, 916 (2018).
17. Agren, R., Mardinoglu, A., Asplund, A., Kampf, C., Uhlen, M. & Nielsen, J. Identification of anti-cancer drugs for hepatocellular carcinoma through personalized genome-scale metabolic modeling. *Molecular systems biology* **10**, 721 (2014).
18. Renz, A., Hohner, M., Breitenbach, M., Josephs-Spaulding, J., Dürrwald, J., Best, L., Jami, R., Marinos, G., Cabreiro, F., Dräger, A., *et al.* Metabolic Modeling Elucidates Phenformin and Atpenin A5 as Broad-Spectrum Antiviral Drugs (2023).
19. Rigauts, C., Aizawa, J., Taylor, S. L., Rogers, G. B., Govaerts, M., Cos, P., Ostyn, L., Sims, S., Vandeplassche, E., Sze, M., *et al.* *Rothia mucilaginosa* is an anti-inflammatory bacterium in the respiratory tract of patients with chronic lung disease. *European Respiratory Journal* **59** (2022).
20. Dastager, S. G., Krishnamurthi, S., Rameshkumar, N. & Dharne, M. The family micrococcaceae. *The Prokaryotes*, 455–498 (2014).
21. Cho, E.-J., Sung, H., Park, S.-J., Kim, M.-N. & Lee, S.-O. *Rothia mucilaginosa* pneumonia diagnosed by quantitative cultures and intracellular organisms of bronchoalveolar lavage in a lymphoma patient. *Annals of Laboratory Medicine* **33**, 145 (2013).
22. Lambotte, O., Debord, T., Soler, C. & Roué, R. Pneumonia due to *Stomatococcus mucilaginosus* in an AIDS patient: case report and literature review. *Clinical microbiology and infection* **5**, 112–114 (1999).
23. Robertson, R. D., Panigrahi, A. & Cheema, R. *Rothia mucilaginosa* bacteremia, meningitis leading to diffuse cerebritis in an adolescent patient undergoing acute myeloid leukemia chemotherapy causing significant morbidity. *SAGE Open Medical Case Reports* **9**, 2050313X211063745 (2021).

24. Sánchez-Carrillo, C., Cercenado, E., Cibrián, F. & Bouza, E. *Stomatococcus mucilaginosus* pneumonia in a liver-transplant patient. *Clinical Microbiology Newsletter* **7**, 54–55 (1995).
25. Hopkins, R. J., Schwalbe, R. S. & Donnenberg, M. Infections due to *Stomatococcus mucilaginosus*: report of two new cases and review. *Clinical infectious diseases* **14**, 1264–1264 (1992).
26. Bergan, T. & Kocur, M. *Stomatococcus mucilaginosus* gen. nov., sp. nov., ep. rev., a member of the family *Micrococcaceae*. *International Journal of Systematic and Evolutionary Microbiology* **32**, 374–377 (1982).
27. Granlund, M., Linderholm, M., Norgren, M., Olofsson, C., Wahlin, A. & Holm, S. E. *Stomatococcus mucilaginosus* septicemia in leukemic patients. *Clinical Microbiology and Infection* **2**, 179–185 (1996).
28. Al-Jebouri, M. & Younis, A. Incidence of aminoglycosides-resistant *Rothia mucilagenosa* causing respiratory infections in workers of Al-Baiji oil refinery, Iraq. (2012).
29. Von Graevenitz, A. *Rothia dentocariosa*: taxonomy and differential diagnosis. *Clinical microbiology and infection* **10**, 399–402 (2004).
30. Machado, D., Andrejev, S., Tramontano, M. & Patil, K. R. Fast automated reconstruction of genome-scale metabolic models for microbial species and communities. *Nucleic acids research* **46**, 7542–7553 (2018).
31. Lieven, C., Beber, M. E., Olivier, B. G., Bergmann, F. T., Ataman, M., Babaei, P., Bartell, J. A., Blank, L. M., Chauhan, S., Correia, K., *et al.* MEMOTE for standardized genome-scale metabolic model testing. *Nature Biotechnology*. ISSN: 1546-1696 (3 Mar. 2020).
32. Chan, S. H., Cai, J., Wang, L., Simons-Senftle, M. N. & Maranas, C. D. Standardizing biomass reactions and ensuring complete mass balance in genome-scale metabolic models. *Bioinformatics* **33**, 3603–3609 (2017).
33. Feierabend, M., Renz, A., Zelle, E., Nöh, K., Wiechert, W. & Dräger, A. High-Quality Genome-Scale Reconstruction of *Corynebacterium glutamicum* ATCC 13032. *Frontiers in Microbiology* **12**, 3432 (2021).
34. Gottstein, W., Olivier, B. G., Bruggeman, F. J. & Teusink, B. Constraint-based stoichiometric modelling from single organisms to microbial communities. *Journal of the Royal Society Interface* **13**, 20160627 (2016).
35. Gao, B., Gallagher, T., Zhang, Y., Elbadawi-Sidhu, M., Lai, Z., Fiehn, O. & Whiteson, K. L. Tracking polymicrobial metabolism in cystic fibrosis airways: *Pseudomonas aeruginosa* metabolism and physiology are influenced by *Rothia mucilagenosa*-derived metabolites. *Msphere* **3**, 10–1128 (2018).
36. Henson, M. A. & Hanly, T. J. Dynamic flux balance analysis for synthetic microbial communities. *IET systems biology* **8**, 214–229 (2014).
37. Lim, Y. W., Schmieder, R., Haynes, M., Furlan, M., Matthews, T. D., Whiteson, K., Poole, S. J., Hayes, C. S., Low, D. A., Maughan, H., *et al.* Mechanistic model of *Rothia mucilagenosa* adaptation toward persistence in the CF lung, based on a genome reconstructed from metagenomic data. *PloS one* **8**, e64285 (2013).
38. Uranga, C. C., Arroyo Jr, P., Duggan, B. M., Gerwick, W. H. & Edlund, A. Commensal oral *Rothia mucilagenosa* produces enterobactin, a metal-chelating siderophore. *Msystems* **5**, 10–1128 (2020).

Re: Spectrum04006-23R1 (Genome-scale model of *Rothia mucilaginosa* predicts gene essentialities and reveals metabolic capabilities)

Dear Ms. Nantia Leonidou:

Your manuscript has been accepted, and I am forwarding it to the ASM production staff for publication. Your paper will first be checked to make sure all elements meet the technical requirements. ASM staff will contact you if anything needs to be revised before copyediting and production can begin. Otherwise, you will be notified when your proofs are ready to be viewed.

Sincerely,
Angela Re
Editor
Microbiology Spectrum

Reviewer #1 (Comments for the Author):

This study thoroughly developed a high-quality, manually curated metabolic model for *R. mucilaginosa*. It highlights the model's properties and its ability to predict growth in various conditions accurately. The gene essentiality analysis is now appropriately contextualized, so the authors have addressed all my questions and concerns.

Reviewer #2 (Comments for the Author):

The authors have successfully addressed all my comments.